# Predictive Preference Learning from Human Interventions

**Haoyuan Cai, Zhenghao Peng, Bolei Zhou**
Department of Computer Science,
University of California, Los Angeles

## Abstract

Learning from human involvement aims to incorporate the human subject to monitor and correct agent behavior errors. Although most interactive imitation learning methods focus on correcting the agent's action at the current state, they do not adjust its actions in future states, which may be potentially more hazardous. To address this, we introduce Predictive Preference Learning from Human Interventions (PPL), which leverages the implicit preference signals contained in human interventions to inform predictions of future rollouts. The key idea of PPL is to bootstrap each human intervention into $L$ future time steps, called the preference horizon, with the assumption that the agent follows the same action and the human makes the same intervention in the preference horizon. By applying preference optimization on these future states, expert corrections are propagated into the safety-critical regions where the agent is expected to explore, significantly improving learning efficiency and reducing human demonstrations needed. We evaluate our approach with experiments on both autonomous driving and robotic manipulation benchmarks and demonstrate its efficiency and generality. Our theoretical analysis further shows that selecting an appropriate preference horizon $L$ balances coverage of risky states with label correctness, thereby bounding the algorithmic optimality gap. Demo and code are available at: `https://metadriverse.github.io/ppl`.

## 1 Introduction

Effectively leveraging human demonstrations to teach and align autonomous agents remains a central challenge in both Reinforcement Learning (RL) [46] and Imitation Learning (IL) [17]. In the literature of RL and more recent RL from Human Feedback (RLHF), the agent explores the environment through trial and error or under human feedback guidance, and the learning process hinges on a carefully crafted reward function that reflects human preferences. However, RL algorithms often require a large number of environment interactions to learn stable policies, and their exploration can lead agents to dangerous or task-irrelevant states [40, 27]. In contrast, IL methods train agents to emulate human behavior using offline demonstrations from experts. Nevertheless, IL agents are susceptible to distributional shift because the offline dataset may lack corrective samples in safety-critical or out-of-distribution states [35, 32, 3, 47].

Interactive Imitation Learning (IIL) [2, 34, 15, 42, 28, 41, 19, 20] incorporates human participants to intervene in the training process and provide online demonstrations. Such methods have improved alignment and learning efficiency in a wide variety of tasks, including robot manipulation [7, 8], autonomous driving [27, 28], and even the strategy game StarCraft II [39]. One line of research on confidence-based IIL designs various task-specific criteria to request human help, including uncertainty estimation [23] and confidence in completing the task [4, 38]. In contrast, an increasing body of work focuses on learning from active human involvement, where human subjects actively intervene and provide demonstrations during training when the agent makes mistakes [15, 42, 22, 17, 28]. Compared to confidence-based IIL, active human involvement can ensure training safety [27]

39th Conference on Neural Information Processing Systems (NeurIPS 2025).

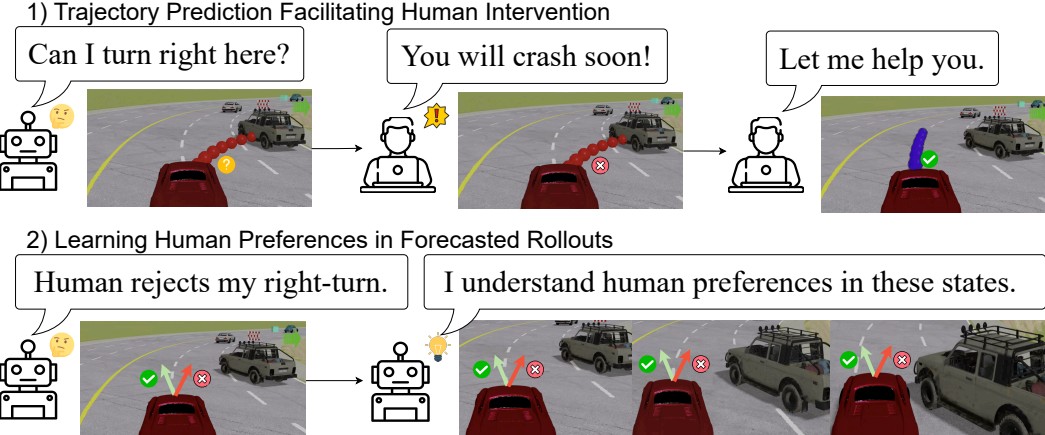

Figure 1: Our Predictive Preference Learning from Human Interventions. (Top) Our approach forecasts the agent's upcoming trajectory (the red dotted path) and visualizes it for the human expert, who will intervene if the forecasted path indicates an upcoming failure. (Bottom) A single intervention is then interpreted as hypothesized preference signals across the predicted states. These signals reflect the agent's imputed imagination of what the expert would prefer, guiding the policy to avoid the risky maneuver in similar future contexts. This integration of proactive forecasting and preference learning accelerates policy improvement and reduces the total number of expert interventions required.

and does not require carefully designing human intervention criteria for each task [12]. However, these methods require the human expert to monitor the entire training process, predict the agent's future trajectories, and intervene immediately in safety-critical states [28], imposing a significant cognitive burden on the human participant. In addition, these methods often correct the agent's behavior only at the current intervened state, penalizing undesired actions step by step. For instance, in HG-DAgger [15], the agent is optimized to mimic human actions solely at the states where interventions occur. In practice, it is intuitive that the agent may repeat similar mistakes in the consecutive future steps $t + 1, \cdots, t + L$ following an error at step $t$. As a result, the expert must repeatedly provide corrective demonstrations in these regions, compromising training efficiency [17].

In this work, we propose a novel Interactive Imitation Learning algorithm, ***Predictive Preference Learning*** from Human Interventions (PPL), to learn from active human involvement. As shown in Fig. 1, our approach has two key designs: First, we employ an efficient rollout-based trajectory prediction model to forecast the agent's future states. These predicted rollouts are visualized in real time for the user, helping human supervisors proactively determine when an intervention is necessary. Second, our algorithm leverages preference learning on the predicted future trajectories to further improve the sample efficiency and reduce the expert demonstrations needed. Such designs bring three strengths: (1) They mitigate the distributional shift problem in IIL and improve training efficiency. By incorporating anticipated future states into the training process, our method constructs a richer dataset, especially in safety-critical situations. This expanded dataset offers more information than expert corrective demonstrations in human-intervened states only. (2) The preference learning reduces the agent's visits to dangerous states, thus suppressing human interventions in safety-critical situations. (3) By visualizing the agent's predicted future trajectories in the user interface, we significantly reduce the cognitive burden on the human supervisor to constantly anticipate the agent's behavior.

Our contributions can be summarized as follows:

1. We introduce a novel Interactive Imitation Learning (IIL) algorithm that leverages trajectory prediction to inform human intervention and employs preference learning to deter the agent from returning to dangerous states.

2. We evaluate our algorithm on the MetaDrive [16] and Robosuite [49] benchmarks, using both neural experts and real human participants, showing that PPL requires fewer expert monitoring efforts and demonstrations to achieve near-optimal policies.

3. We present a theoretical analysis that derives an upper bound on the performance gap of our approach. This bound highlights that the efficacy of our method lies in reducing distributional shifts while preserving the quality of preference data.

## 2   Related Work

**Learning from Human Involvement.**   Many works incorporate human involvement in the training loop to provide corrective actions in dangerous or repetitive states. For example, Human-Gated DAgger (HG-DAgger) [15], Ensemble-DAgger [23], Thrifty-DAgger [12], Sirius [19], and Intervention Weighted Regression (IWR) [22] perform imitation learning on human intervention data. These methods do not leverage data collected by agents or suppress undesired actions likely to be intervened by humans, leading to the agent's susceptibility to entering hazardous states and thus harming sample efficiency. EGPO [27], PVP [28], and AIM [2] design proxy cost or value functions to suppress the frequency of human involvement. However, these approaches still require human supervisors to continuously monitor the agent's behavior throughout training and anticipate potential failures that may necessitate intervention. This continuous oversight imposes a significant cognitive load on the human expert and can limit scalability. Furthermore, these methods do not exploit the agent's predicted future trajectories that the expert might identify as potentially leading to undesirable outcomes, which necessitates repeated corrective demonstrations in such situations.

**Preference-Based RL.**   A large body of work focuses on learning human preferences by ranking pairs of trajectories generated by the agent [6, 9, 34, 44, 37, 26]. One prominent paradigm, reinforcement learning from human feedback (RLHF), first trains a reward model on offline human preference data and then uses that model to guide policy optimization [6, 25, 43]. RLHF has achieved impressive results in domains ranging from Atari games [6] to large language models [25]. Alternatively, methods such as Direct Preference Optimization (DPO) [31], Contrastive Preference Optimization (CPO) [45], and related variants [1, 24] bypass explicit reward-model training and instead directly optimize the policy to satisfy preference labels via a classification loss.

However, applying RLHF and DPO to real-time control problems faces challenges due to the need for extensive human labeling of preference data [31]. These labels are inherently subjective and prone to noise [37]. Moreover, acquiring a high-quality preference dataset and achieving near-optimal policies often requires a substantial number of environment samples, thereby imposing a considerable burden on human experts [10]. In contrast, our framework elicits preferences in an online, interactive manner: experts review the agent's predicted future trajectory at each decision point and intervene when a failure is anticipated; these interventions are then converted into contrastive preference labels. This real-time preference collection enables the policy to adapt continuously to the evolving state distribution and to receive targeted feedback precisely where it is most needed. In summary, our approach PPL bridges preference-based RL and imitation learning by demonstrating that DPO-style alignment techniques can be effectively adapted to control problems within an interactive imitation learning framework.

## 3   Problem Formulation

In this section, we introduce our settings of interactive imitation learning environments. We use the Markov decision process (MDP) $M = \langle \mathcal{S}, \mathcal{A}, \mathcal{P}, r, \gamma, d_0 \rangle$ to model the environment, which contains a state space $\mathcal{S}$, an action space $\mathcal{A}$, a state transition function $\mathcal{P} : \mathcal{S} \times \mathcal{A} \to \mathcal{S}$, a reward function $r : \mathcal{S} \times \mathcal{A} \to [R_{\min}, R_{\max}]$, a discount factor $\gamma \in (0, 1)$, and an initial state distribution $d_0 : \mathcal{S} \to [0, 1]$. We denote $\pi(a \mid s) : \mathcal{S} \times \mathcal{A} \to [0, 1]$ as a stochastic policy. Reinforcement learning (RL) aims to learn a *novice policy* $\pi_n(a|s)$ that maximizes the expected cumulative return $J(\pi_n) = \mathbb{E}_{\tau \sim P_{\pi_n}} [\sum_{t=0}^{\infty} \gamma^t r(s_t, a_t)]$, wherein $\tau = (s_0, a_0, s_1, a_1, ...)$ is the trajectory sampled from trajectory distribution $P_{\pi_n}$ induced by $\pi_n$, $d_0$ and $\mathcal{P}$. We also define the discounted state distribution under policy $\pi_n$ as $d_{\pi_n}(s) = (1-\gamma) \mathbb{E}_{\tau \sim P_{\pi_n}} [\sum_{t=0}^{\infty} \gamma^t \mathbb{I}[s_t = s]]$. In this work, we consider the reward-free setting where the agent has no access to the task reward function $r(s, a)$.

In imitation learning (IL), we assume that the human expert behavior $a_h$ follows a *human policy* $\pi_h(a \mid s)$. The agent aims to learn $\pi_n$ from human expert trajectories $\tau_h \sim P_{\pi_h}$, and it needs to optimize $\pi_n$ to close the gap between $\tau_n \sim P_{\pi_n}$ and $\tau_h$. Prior works on imitation learning have shown that using an offline expert demonstration dataset may lead to poor performance due to out-of-distribution states [36, 34, 41]. Therefore, interactive imitation learning (IIL) methods incorporate a human expert into the training loop to provide online corrective demonstrations, making

the state distribution of expert data more similar to that of the novice policy [28, 35]. During training, the human expert monitors the agent and can intervene and take control if the agent's action $a_n$ at the current state $s$ violates the human's desired behavior or leads to a dangerous situation. We use the deterministic intervention policy $I(s, a_n) : \mathcal{S} \times \mathcal{A} \to \{0, 1\}$ to model the human's intervention behavior, where the agent's action follows the novice policy $a_n \sim \pi_n(\cdot \mid s)$, and the human subject takes control when $I(s, a_n) = 1$.

With the notations above, the agent's actual trajectories during training are derived from the following shared *behavior policy*

$$\pi_b(a \mid s) = \pi_n(a \mid s)(1 - I(s, a)) + \pi_h(a \mid s)G(s), \tag{1}$$

wherein $G(s) = \int_{a' \in \mathcal{A}} I(s, a')\pi_n(a' \mid s)da'$ is the probability of the agent taking an action $a_n$ that will be rejected and intervened by the human expert.

**Preference Alignment.** Recent works on preference-based RL have also leveraged offline preference datasets to learn human-aligned policies [31, 45, 1]. Given an offline preference dataset $\mathcal{D}_{\text{pref}}$ where each preference data $(s, a^+, a^-) \in \mathcal{D}_{\text{pref}}$ means that the human expert prefers the action $a^+$ over $a^-$ at state $s$, we can learn an agent policy $\pi_n$ that aligns with the human preference model. The Contrastive Preference Optimization method [45] uses the following objective to train an agent policy $\pi_\theta$ from the preference dataset $\mathcal{D}_{\text{pref}}$:

$$\mathcal{L}_{\text{pref}}(\pi_\theta) = - \mathop{\mathbb{E}}_{(s, a^+, a^-) \sim \mathcal{D}_{\text{pref}}} \left[ \log \sigma \left( \beta \log \pi_\theta(a^+ \mid s) - \beta \log \pi_\theta(a^- \mid s) \right) \right], \tag{2}$$

where $\sigma(\cdot)$ is the Sigmoid function, and $\beta > 0$ is a hyperparameter.

**Trajectory Prediction Model.** In this work, we allow the agent to access a short-term trajectory prediction model $f(s, a_n, H)$. Given the current state $s$ and the agent's action $a_n$, we can predict the agent's trajectory $f(s, a_n, H) = (s, \tilde{s}_1, \cdots, \tilde{s}_H)$ in the next $H$ steps, where $\tilde{s}_i$ the predicted state that the agent will reach if the agent applies the action $a_n$ for $i$ steps from the state $s$. The implementation detail of $f$ is in Sec. 4.3.

## 4 Method

### 4.1 Predictive Preference Learning from Human Interventions (PPL)

We propose PPL (Fig. 2), an efficient interactive imitation learning method that emulates the human policy with fewer expert demonstrations and less cognitive effort. The key idea of PPL is to learn human preferences from data generated by a future-trajectory prediction model. We illustrate the human-agent interactions in Fig. 2 (left) and how PPL infers human preference in Fig. 2 (right).

During training, the human subject monitors the agent-environment interaction in each state $s$ (Fig. 2 (left)). The novice policy $\pi_n$ suggests an action $a_n$ for the current state $s$. Instead of executing $a_n$ immediately, we query the trajectory prediction model $f(s, a_n, H)$ to obtain a predicted rollout $\tau = f(s, a_n, H) = (s, \tilde{s}_1, \cdots, \tilde{s}_H)$, which we visualize for the human expert. The expert then uses $\tau$ to determine whether the agent will fail in the next $H$ steps, such as crashing into vehicles or going off the road. If so, the expert will provide corrective actions $a_h \sim \pi_h(s)$ for the next $H$ steps, depicted by the blue trajectory in Fig. 2. If the expert believes no intervention is needed, the agent continues to use its own policy $\pi_n$ for the next $H$ steps.

We introduce preference learning on the predicted trajectories because it is difficult to learn corrective behavior purely from the expert's demonstrations in safe states. By visualizing predicted rollouts, experts can anticipate unsafe trajectories before the agent actually enters them and intervene preemptively. As a result, the state distribution covered by these early interventions differs substantially from the on-policy distribution of the novice policy, creating a distributional shift that standard imitation or on-policy correction cannot address. Therefore, instead of relying solely on expert demonstrations, we collect preference labels over the predicted rollouts (Fig. 2 Right) so that the agent can learn the correct behavior in those risky states.

Whenever the expert intervenes at state $s$, we interpret this as indicating that continuing with $a_n$ would lead to unsafe or undesirable outcomes along the predicted trajectory. As shown in Fig. 2 (right), to capture this preference, we assume the expert prefers $a_h$ over $a_n$ at state $s$ and each of

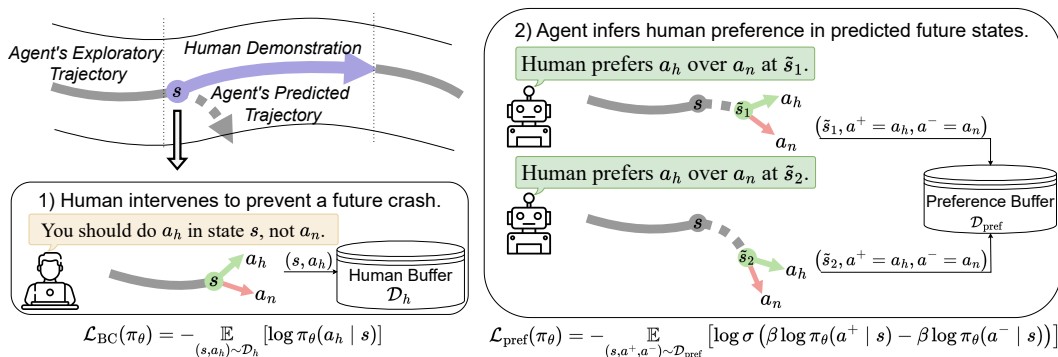

$$\mathcal{L}_{\text{BC}}(\pi_\theta) = - \underset{(s,a_h) \sim \mathcal{D}_h}{\mathbb{E}} \left[ \log \pi_\theta(a_h \mid s) \right] \qquad \mathcal{L}_{\text{pref}}(\pi_\theta) = - \underset{(s,a^+,a^-) \sim \mathcal{D}_{\text{pref}}}{\mathbb{E}} \left[ \log \sigma \left( \beta \log \pi_\theta(a^+ \mid s) - \beta \log \pi_\theta(a^- \mid s) \right) \right]$$

Figure 2: Illustration of Predictive Preference Learning. (Left) At each decision point, the agent proposes an action, and its future trajectory is predicted and visualized. The human expert reviews this rollout and intervenes only when a potential failure is anticipated. The intervention is recorded alongside the state into the human buffer $\mathcal{D}_h$ for behavioral cloning. (Right) Each recorded intervention is then converted into contrastive preference pairs over the predicted future states $\tilde{s}_1, \cdots, \tilde{s}_L$. These preference tuples are stored in a preference buffer $\mathcal{D}_{\text{pref}}$ and used to train the policy via a contrastive classification loss, propagating expert intents into regions the agent is likely to explore.

the first $L$ predicted states $\tilde{s}_1, \cdots, \tilde{s}_L$ for some preference horizon $L \leq H$. For each $i \leq L$, we add the tuple $(\tilde{s}_i, a^+ = a_h, a^- = a_n)$ to the preference dataset $\mathcal{D}_{\text{pref}}$. We note that in each tuple $(\tilde{s}_i, a^+ = a_h, a^- = a_n)$, both $a_h$ and $a_n$ are sampled at the current state $s$, not the predicted future states $\tilde{s}_i$, because the exact human corrective actions at hypothetical future states are not directly observable. Still, the expert intervention at state $s$ implies that applying $a_h$ at the predicted states $\tilde{s}_1, \ldots, \tilde{s}_L$, rather than continuing with $a_n$, is more likely to prevent the dangerous outcome in the end of the predicted trajectory ($\tilde{s}_H$). Hence, our construction of the preference dataset ensures that it faithfully captures the expert's corrective intent across the predicted horizon.

The preference horizon $L$ controls the length over which we elicit preferences in the predicted trajectory. A small $L$ may fail to capture enough risky states, while a large $L$ risks applying preferences where the corrective action $a_h$ at state $s$ no longer matches what an expert would do in those imagined states $\tilde{s}_i$. In Theorem 4.1, we prove that under mild assumptions, the performance gap of our learned policy is bounded by terms reflecting the state distribution shift and the quality of the preference labels, implying that an ideal preference horizon $L$ should balance these two error terms. We also illustrate how the choice of $L$ affects the performance of PPL in Fig. 8.

We train the novice policy $\pi_n$ using two complementary objectives. First, we apply a behavioral cloning loss on expert demonstrations $\mathcal{D}_h$:

$$\mathcal{L}_{\text{BC}}(\pi_\theta) = - \underset{(s,a_h) \sim \mathcal{D}_h}{\mathbb{E}} \left[ \log \pi_\theta(a_h \mid s) \right]. \qquad (3)$$

Second, inspired by Contrastive Preference Optimization (CPO) [45], we use the preference-classification loss Eq. 2 over the predicted states in $\mathcal{D}_{\text{pref}}$. The final loss of the agent policy $\pi_\theta$ is evaluated as

$$
\begin{aligned}
\mathcal{L}(\pi_\theta) &= \mathcal{L}_{\text{pref}}(\pi_\theta) + \mathcal{L}_{\text{BC}}(\pi_\theta) \\
&= - \underset{(s,a^+,a^-) \sim \mathcal{D}_{\text{pref}}}{\mathbb{E}} \left[ \log \sigma \left( \beta \log \pi_\theta(a^+ \mid s) - \beta \log \pi_\theta(a^- \mid s) \right) \right] - \underset{(s,a_h) \sim \mathcal{D}_h}{\mathbb{E}} \left[ \log \pi_\theta(a_h \mid s) \right].
\end{aligned}
\qquad (4)
$$

The workflow of our method PPL is summarized in Alg. 1.

## 4.2 Analysis

We prove that the performance gap between the human policy $\pi_h$ and the agent policy $\pi_n$ can be bounded by the following three error terms: 1) the state distribution shift $\delta_{\text{dist}}$, 2) the quality of the preference labels $\delta_{\text{pref}}$, and 3) the optimization error $\epsilon$.

The first error term is defined as $\delta_{\text{dist}} = D_{\text{TV}}(d_{\pi_n}, d_{\text{pref}})$, where $d_{\pi_n}(s)$ is the discounted state distribution of the agent's policy $\pi_n$, and $d_{\text{pref}}(s) = |\mathcal{D}_{\text{pref}}|^{-1} \mathbb{E}_{(s',a^+,a^-) \sim \mathcal{D}_{\text{pref}}} \mathbb{I}[s' = s]$. Here,

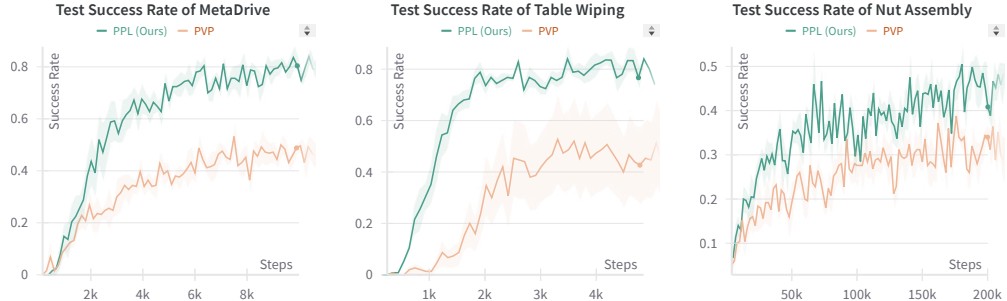

Figure 3: The test-time performance curve of PPL and the IIL counterpart PVP [28] under three different environments. The x-coordinate is the number of environment interactions, and the y-coordinate is the agent's success rate in a held-out test environment, where the evaluation is conducted without expert involvement. Compared to the IIL counterpart, our approach achieves much higher learning efficiency and reduces the expert's efforts needed.

$D_{\text{TV}}(P, Q) = \frac{1}{2}|P - Q|_1$ is the total variation distance between two distributions. This error term quantitatively measures the difference between the states actually visited by the agent and those contained in the preference dataset.

The second error term is defined as $\delta_{\text{pref}} = \underset{s \sim d_{\text{pref}}}{\mathbb{E}} D_{\text{TV}}(\rho^s_{\text{ideal}}, \rho^s_{\text{pref}})$, which arises from the misalignment of the positive actions in the preference dataset, as the human action $a_h$ in each tuple $(\tilde{s}_i, a_h, a_n) \in \mathcal{D}_{\text{pref}}$ is sampled in state $s$ instead of state $\tilde{s}_i$. That is, this error reflects the assumption that the human would still apply the same corrective action $a_h$ in a hypothetical future state $\tilde{s}_i$ reached after executing $a_n$ for $i$ steps, which may not perfectly match what the expert would actually do. For any state $s$ in $\mathcal{D}_{\text{pref}}$ with $d_{\text{pref}}(s) > 0$, the empirical preference-pair distribution in state $s$ follows

$$\rho^s_{\text{pref}}(a_h, a_n) = \frac{\mathbb{E}_{(s', a^+, a^-) \sim \mathcal{D}_{\text{pref}}} \mathbb{I}[s' = s, a^+ = a_h, a^- = a_n]}{\mathbb{E}_{(s', a^+, a^-) \sim \mathcal{D}_{\text{pref}}} \mathbb{I}[s' = s]}. \tag{5}$$

The ideal preference-pair distribution at any state $s$ in $\mathcal{D}_{\text{pref}}$ is simply the joint distribution of $(a_h, a_n)$: $\rho^s_{\text{ideal}}(a_h, a_n) = \pi_h(a_h \mid s)\pi_n(a_n \mid s)$ on $\mathcal{A} \times \mathcal{A}$.

Finally, we define the optimization error of the agent policy $\pi_n$ as $\epsilon = \mathcal{L}_{\text{pref}}(\pi_n) - \mathcal{L}_{\text{pref}}(\pi_h)$. We recall that $\mathcal{L}_{\text{pref}}(\pi) = - \underset{(s, a^+, a^-) \sim \mathcal{D}_{\text{pref}}}{\mathbb{E}} [\log \sigma (\beta \log \pi(a^+ \mid s) - \beta \log \pi(a^- \mid s))]$, where $\beta$ is a positive constant and $\sigma(\cdot)$ is the Sigmoid function. Under these notations, we have the following Thm. 4.1.

**Theorem 4.1.** *We denote the Q-function of the human policy $\pi_h$ as $Q^*(s, a)$. We assume that for any $(s, a, a')$, $|Q^*(s, a) - Q^*(s, a')| \leq U$, $|\log \pi_h(a|s) - \log \pi_h(a'|s)| \leq M$, and $|\log \pi_n(a|s) - \log \pi_n(a'|s)| \leq M$, where $U, M > 0$ are constants. When $\beta$ is small enough, we have*

$$J(\pi_h) - J(\pi_n) = O(\sqrt{\epsilon + \delta_{\text{pref}}} + \delta_{\text{dist}}). \tag{6}$$

Here we explain the insights of Thm. 4.1 as follows. In our choice of the preference horizon $L$, the key is to balance the two error terms $\delta_{\text{dist}}$ and $\delta_{\text{pref}}$. Recall that the distribution shift term $\delta_{\text{dist}}$ measures how close the state distributions are when there is no human intervention ($d_{\pi_n}$) and the state distribution represented in the preference dataset ($d_{\text{pref}}$). Increasing $L$ decreases $\delta_{\text{dist}}$ because the preference dataset will contain more predicted states $\tilde{s}_i$ from the agent's future trajectories. In contrast, the preference error term $\delta_{\text{pref}}$ captures the misalignment between the true but unobserved human action at a future state $t' > t$ and the bootstrapped corrective action $a_h$ at step $t$, which we assume would also apply at $t'$. Therefore, the longer the preference horizon, the larger $\delta_{\text{pref}}$, because the difference between the human actions $a_h$ in state $s$ and the predicted $\tilde{s}_L$ grows as $L$ increases. In Fig. 8, we visualize the effects of $L$ on the performance of PPL. We prove Thm. 4.1 in Appendix F.

### 4.3 Implementation Details

**Tasks.** As shown in Fig. 4, we conduct experiments on control tasks and manipulation tasks with different observation and action spaces. For the control task, we consider the MetaDrive driving

experiments [16], where the agent must navigate towards the destination in heavy-traffic scenes without crashing into obstacles or other vehicles. The agent uses the sensory state vector $s \in \mathbb{R}^{259}$ as its observation and outputs a control signal $a = (a_0, a_1) \in [-1, 1]^2$ representing the steering angle and the acceleration, respectively. We evaluate the agent's learned policy in a held-out test environment separate from the training environments.

For manipulation tasks, we consider the Table Wiping and Nut Assembly tasks from the Robosuite environment [49]. In the Table Wiping task, the robot arm must learn to wipe the whiteboard surface and clean all of the markings. The positions of these markings are randomized at the beginning of each episode. The states are $s \in \mathbb{R}^{34}$ and actions are $a \in \mathbb{R}^6$ (3 translations in the XYZ axes and 3 rotations around the XYZ axes). In the Nut Assembly task, the robot must grab a metal ring from a random initial pose and place it over a target cylinder at a fixed location. The states are $s \in \mathbb{R}^{51}$ and actions are $a \in \mathbb{R}^7$, where the additional dimension in the action space represents opening or closing the gripper. In both manipulation tasks, the simulated UR5e robot arm uses fixed-impedance operational-space control to achieve the commanded pose.

**Trajectory Prediction Model.** In PPL, we need to predict the future states $f(s, a_n, H) = (s, \tilde{s}_{t+1}, \cdots, \tilde{s}_{t+H})$ from the current state $s$. We implement $f$ by running an $H$-step simulator rollout from the current state $s$, repeatedly applying action $a_n$ to collect the sequence $(\tilde{s}_{t+1}, \cdots, \tilde{s}_{t+H})$. This $H$-step simulator rollout runs at up to 1,000 fps on a CPU.

In real-world tasks such as autonomous driving, simulator rollouts often deviate from reality because vehicle dynamics parameters are imperfect and other traffic participants behave unpredictably. To predict future motion with minimal overhead, prior work directly propagates the ego-vehicle's state through a physics model [18, 29, 13]. Following this approach, we use the kinematic bicycle model [30] to simulate $H = 10$ steps, assuming all other traffic participants remain stationary. Compared with the data-driven approaches [50, 5, 21], this rule-based predictor requires only forward integration of a single vehicle and produces short-term trajectories whose accuracy closely matches simulator rollouts. This lightweight extrapolation method runs at about 3,000 fps on a CPU, enabling real-time prediction with minimal overhead. Our ablation studies confirm that replacing the simulator with our bicycle-model predictions incurs negligible performance loss (Table 2, rows 9-10).

## 5 Experiments

### 5.1 Experimental Setting

**Neural Policies as Proxy Human Policies.** Experiments with real human participants are time-consuming and exhibit high variability between trials. Following the prior works on interactive imitation learning [10, 27], in addition to real-human experiments, we also incorporate neural policies in the training loop of PPL to approximate human policies in Table 3, 4, and 5. The neural experts are trained using PPO-Lagrangian [33] for 20 million environment steps.

In MetaDrive, the neural expert uses the following takeover rule when training all baselines and our method PPL: if the predicted trajectory $\tau = f(s, a_n, H)$ contains any safety violation, such as crashes or going off the road, or the average speed is too slow, the expert takes control for the next $H$ steps. In RoboSuite, the neural expert intervenes when the cumulative reward over the predicted trajectory $\tau$ falls below a threshold $\epsilon$. We set $\epsilon = 1$ for the Table Wiping task and $\epsilon = 2$ for the Nut Assembly task.

In Table 1, we report experiments involving real humans in the MetaDrive safety benchmark. In Table 3, Table 4, and Table 5, we report experiments with the neural policy as the proxy human policy in the MetaDrive, Table Wiping, and Nut Assembly tasks, respectively.

**Evaluation Metrics.** In the Table Wiping task and Nut Assembly task, we report the *success rate*, the ratio of episodes where the agent reaches the destination. In the MetaDrive safety benchmark, we also report the *episodic return* and *route completion rate* during evaluation. The route completion rate is the ratio of the agent's successfully traveled distance to the length of the complete route.

We train each interactive imitation learning baseline five times using distinct random seeds. Then, we roll out 50 trajectories generated by each model in the held-out evaluation environment and average each evaluation metric as the model's performance. During the evaluation, no expert is involved. The standard deviation is provided. We fix $H = 10$ for all the interactive imitation learning baselines. In

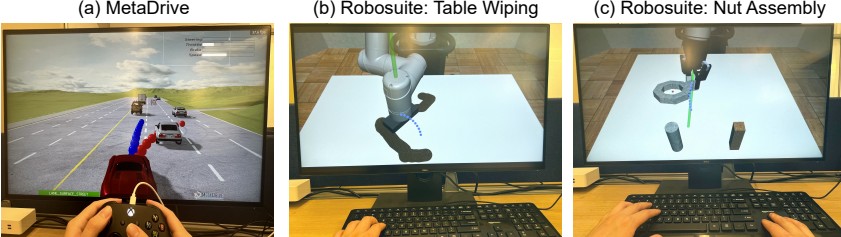

| (a) MetaDrive | (b) Robosuite: Table Wiping | (c) Robosuite: Nut Assembly |

Figure 4: Human interfaces of the three tasks: MetaDrive (a), Table Wiping (b), and Nut Assembly (c). In (a), the agent's forecasted trajectory (the red dots) leads to a collision, prompting the expert to intervene via the gamepad (blue dots show the predicted rollout of the expert). In (b) and (c), the expert observes the agent's forecasted trajectory and intervenes via the keyboard if necessary.

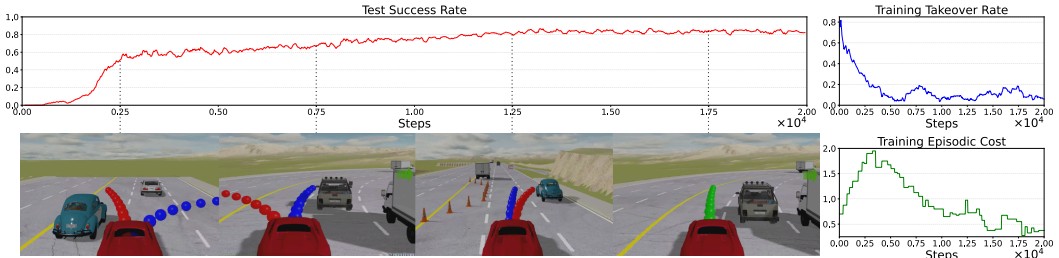

Figure 5: Training process of PPL in the MetaDrive environment with the human expert over 20K steps. We plot the test success rate (left), training takeover rate (top right), and training episodic safety cost (bottom right). During training, when the agent's forecasted trajectory (red dots) leads to a collision, the human expert intervenes via the gamepad, and the corrected rollout is shown (blue dots). When the agent's forecasted trajectory is safe, it is visualized in green dots. The agent becomes autonomous and performant during training, requiring fewer human interventions to maintain safety.

PPL, we fix $\beta = 0.1$, choose $L = 4$ for the MetaDrive benchmark and Table Wiping task, and set $L = 6$ for the Nut Assembly task. In Fig. 8, we show how the choice of $L$ affects the performance of PPL in the MetaDrive benchmark.

We also report the total number of human-involved transitions (*human data usage*) and the *overall intervention rate*, which is the ratio of human data usage to total data usage. These show how much effort humans make to teach the agents.

**Human Interfaces.** Human subjects can take control through the Xbox Wireless Controller or the keyboard and monitor the training process by visualizing environments on the screen. The predicted trajectories are updated every $H = 10$ steps (one second), so that the human expert can intervene promptly before the agent causes any safety violations and undesired behaviors.

**Baselines.** We test two imitation learning baselines: Behavior Cloning (BC) and GAIL [11], and two confidence-based IIL methods: Ensemble-DAgger [23] and Thrifty-DAgger [12]. Four human-in-the-loop IIL methods that learn from active human involvement are tested: Intervention Weighted Regression (IWR) [22], Human-AI Copilot Optimization (HACO) [17], Expert Intervention Learning (EIL) [42], and Proxy Value Propagation [28].

## 5.2 Baseline Comparison

In Table 1, we report the performance of our PPL and all the baselines with real human experts in the MetaDrive safety benchmark. Our method PPL outperforms all the baselines and achieves a success rate of 76% within 10K steps. The whole experiment of PPL takes only 12 minutes on a desktop computer with an Nvidia GeForce RTX 4080 GPU.

In Table 3, 4, and 5, we report the performance of our PPL and all the baselines with neural experts as proxy human policies in MetaDrive, Table Wiping, and Nut Assembly tasks, respectively. We also plot the curves of the test-time success rate in Fig. 3. These tables and Fig. 3 show that PPL achieves both fewer expert data usage and environment samples needed in both driving tasks and robot manipulation tasks while significantly outperforming baselines in testing performance. These

Table 1: Comparison of methods with training/testing statistics in the MetaDrive environment with the real human expert. The overall intervention rate is given together with the human data usage.

| Method | Human-in-the-Loop | Training | | Testing | | |
|---|---|---|---|---|---|---|
| | | Human Data Usage | Total Data Usage | Success Rate | Episodic Return | Route Completion |
| Human Expert | – | 20K | – | $0.95_{\pm 0.04}$ | $349.2_{\pm 18.2}$ | $0.98_{\pm 0.01}$ |
| BC | ✗ | 20K | – | $0.0_{\pm 0.0}$ | $53.5_{\pm 22.8}$ | $0.16_{\pm 0.07}$ |
| GAIL | ✗ | 20K | 1M | $0.14_{\pm 0.03}$ | $146.2_{\pm 17.1}$ | $0.44_{\pm 0.05}$ |
| Ensemble-DAgger | ✓ | 3.8K (0.38) | 10K | $0.36_{\pm 0.11}$ | $233.8_{\pm 21.3}$ | $0.70_{\pm 0.02}$ |
| Thrifty-DAgger | ✓ | 3.2K (0.32) | 10K | $0.45_{\pm 0.04}$ | $221.5_{\pm 26.4}$ | $0.62_{\pm 0.04}$ |
| PVP | ✓ | 4.9K (0.49) | 10K | $0.46_{\pm 0.08}$ | $267.3_{\pm 15.0}$ | $0.71_{\pm 0.04}$ |
| IWR | ✓ | 5.2K (0.52) | 10K | $0.23_{\pm 0.10}$ | $246.7_{\pm 10.7}$ | $0.62_{\pm 0.02}$ |
| EIL | ✓ | 6.9K (0.69) | 10K | $0.01_{\pm 0.01}$ | $137.3_{\pm 26.1}$ | $0.40_{\pm 0.08}$ |
| HACO | ✓ | 6.3K (0.63) | 10K | $0.11_{\pm 0.05}$ | $154.7_{\pm 14.7}$ | $0.45_{\pm 0.09}$ |
| PPL (Ours) | ✓ | **2.9K** (0.29) | 10K | **0.76**$_{\pm 0.07}$ | **324.8**$_{\pm 9.2}$ | **0.90**$_{\pm 0.06}$ |

Table 2: Ablation studies in MetaDrive with 10K total data usage. We use the neural expert as the proxy human policy.

| Method | Expert Data Usage | Route Completion | Success Rate |
|---|---|---|---|
| Imitation on $a^+$ | 1.9K | 0.65 | 0.36 |
| PPL with random $a^+$ | 2.2K | 0.73 | 0.45 |
| PPL with random $a^-$ | 2.3K | 0.69 | 0.38 |
| PPL with DPO | **1.6K** | **0.91** | **0.80** |
| PPL with IPO | 2.6K | 0.61 | 0.35 |
| PPL with SLiC-HF | 3.0K | 0.59 | 0.32 |
| PPL with BC loss only | 2.0K | 0.72 | 0.42 |
| PPL with CPO loss only | 5.8K | 0.31 | 0.04 |
| PPL with rule-based $f$ | **1.9K** | **0.91** | **0.78** |
| PPL (Ours) | **1.8K** | **0.92** | **0.81** |

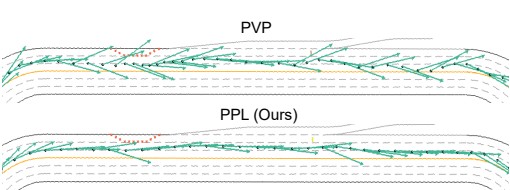

Figure 6: We plot the steering control sequences for both PVP and PPL on the same MetaDrive map, with arrows representing the steering angles every five steps. Both agents are trained to 10K steps. Compared to PVP, our method yields smoother steering and more consistent speeds, especially when navigating close to obstacles.

results suggest that our construction of the preference dataset accurately reflects human preferences and helps speed up imitation learning. In addition, Fig. 6 shows that our method PPL produces smoother control sequences and generates trajectories that better align with human preferences.

## 5.3 Ablation Studies

In Table 2, we perform ablation studies of our PPL in the MetaDrive safety benchmark with the neural expert as proxy human policies.

**Discarding positive or negative actions:** In the first three rows of Table 2, we show that the advantage of our method PPL arises from the constructed preference pairs $(\tilde{s}, a^+, a^-)$ in the preference data $\mathcal{D}_{\text{pref}}$ (Fig. 2 (right)), instead of merely emulating the positive actions $a^+$ or simply avoiding taking the negative actions $a^-$ in the preference buffer. As shown in Table 2, discarding the negative actions $a^-$ and performing Behavior Cloning on the positive actions (Imitation on $a^+$) leads to poor performance, which is even worse than directly imitating the expert demonstrations in the human buffer $\mathcal{D}_h$ (PPL with BC loss only). In addition, replacing the positive actions by random actions (PPL with random $a^+$) or the negative actions by random actions (PPL with random $a^-$) also fails to solve the MetaDrive benchmark.

**Preference-based RL objectives:** In our learning objective Eq. 4, we use the Contrastive Preference Optimization (CPO) loss [45] to learn from the preference dataset $\mathcal{D}_{\text{pref}}$. In Table 2 (rows 4–6), we also report the performance of using other preference-based RL objectives from Direct Preference Optimization (DPO) [31], IPO [1], and SLiC-HF [48]. For DPO and IPO, we use a reference policy trained by Behavior Cloning from 10K expert demonstrations. Table 2 shows that using IPO (PPL with IPO) and SLiC-HF (PPL with SLiC-HF) objectives degrade the performance of PPL. Using the DPO objective (PPL with DPO) does not hurt the performance of PPL. However, the DPO objective requires access to a pretrained reference policy, while our learning objective Eq. 4 does not.

**Discarding the BC loss or preference loss:** As shown in row 7 of Table 2, discarding the CPO loss $\mathcal{L}_{\text{pref}}$ in Eq. 4 (PPL with BC loss only) significantly damages the performance of PPL. Discarding the BC loss (PPL with CPO loss only) also damages the performance, because the BC loss helps regularize our learned policy and avoid it deviating too much from the expert demonstrations.

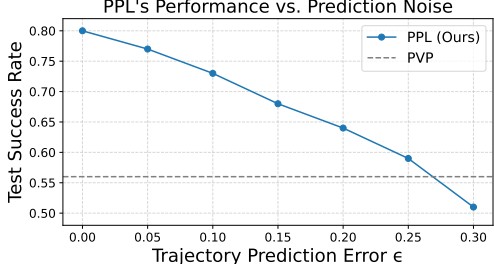
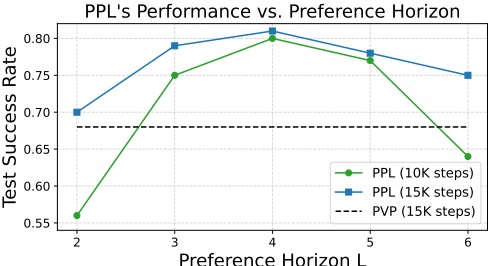

Figure 7: Performance of PPL under varying trajectory-prediction noise levels $\epsilon$ in MetaDrive. PPL still outperforms PVP when the trajectory predictor is imperfect.

Figure 8: Performance of PPL with different preference horizons $L$ in MetaDrive with 10K and 15K total data usage. PPL has the best learning efficiency when we set $L = 4$ in MetaDrive.

**Rule-based trajectory prediction model:** Following Sec. 4.3, we also implement a rule-based trajectory prediction model $f$ by simulating the ego-vehicle dynamics for $H$ steps. Using a rule-based $f$ (PPL with rule-based $f$) has negligible effects on the performance of PPL. This shows that our method still outperforms the IIL baselines even without relying on simulator rollouts.

### 5.4 Robustness Analysis

In Sec. 5.4, we evaluate PPL's robustness to noise in the trajectory predictor (Fig. 7). We also visualize the effect of the preference horizon $L$ on PPL in Fig. 8.

In Fig. 7, we show that PPL is robust to noise in trajectory predictors. With an imperfect predictive model, PPL still outperforms all the baselines. We inject random Gaussian noise $e_{\text{noise}}$ to the outputs $\tilde{s}$ of the trajectory predictor, and we set the norm $||e_{\text{noise}}||_2 = \epsilon * ||\tilde{s}||_2$. Then we gradually increase the constant $\epsilon$ to test PPL's robustness to noises in trajectory predictors. We use MetaDrive, Table Wiping, and Nut Assembly environments following the same setups from Tables 3, 4, and 5, respectively. We find that with a noisy predictive model, PPL still outperforms all the baselines in MetaDrive and Table Wiping when the noise $\epsilon \le 0.25$. In Nut Assembly, PPL outperforms the baselines when $\epsilon \le 0.125$.

In Fig. 8, we visualize how the preference horizon $L$ affects the test success rate of PPL in the MetaDrive safety benchmark with 10K and 15K total data usage. As $L$ increases from 2 to 4, the agent gains additional corrective information from forecasted states in the preference buffer and achieves higher success rates. Beyond $L = 4$, however, the benefit tapers off and eventually degrades, since overly long horizons yield less accurate preference labels. Therefore, we observe peak learning efficiency at $L = 4$. Notably, when $3 \le L \le 5$, PPL trained for only 10K steps already outperforms PVP trained for 15K steps. With an appropriately chosen preference horizon, PPL can substantially reduce both training time and expert monitoring effort.

## 6 Conclusion

In this work, we propose Predictive Preference Learning from Human Interventions (PPL), a novel interactive imitation learning algorithm that applies preference learning over predicted future trajectories to capture implicit human preferences. By converting each expert intervention into contrastive preference labels across forecasted states, PPL directs corrective feedback toward the regions of the state space the agent is most likely to explore. This approach substantially improves learning efficiency and reduces both the number of required demonstrations and the expert's cognitive load, without offline pretraining and reward engineering.

**Limitations.** We assume that the expert always knows the optimal corrective action and demonstrates it accurately, whereas human demonstrations can be suboptimal or inconsistent. Additionally, all our experiments are conducted in simulation. The effectiveness and safety of PPL on real robots operating in physical environments remain to be explored in future works.

**Acknowledgment**: This work was supported by the NSF Grants CCF-2344955 and IIS-2339769, and ONR grant N000142512166. The human experiment in this study is approved through the IRB#23-000116 at UCLA. ZP was supported by the Amazon Fellowship via UCLA Science Hub.

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

# A  Algorithm

We summarize our method PPL in Alg. 1.

---

**Algorithm 1** Predictive Preference Learning from Human Interventions (PPL)

---

1: **Input:** Hyperparameters $H, L, \beta$.
2: **for** timestep $k = 0, H, 2H, \ldots$ **do**
3:     Agent samples action $a_n \sim \pi_n(s_k)$.
4:     Predict future trajectory $\tau = f(s_k, a_n, H) = (s_k, \tilde{s}_{k+1}, \cdots, \tilde{s}_{k+H})$.
5:     Human observes $\tau$ to decide whether to take over in the next $H$ steps.
6:     **for** timestep $t = k, k+1, \cdots, k+H-1$ **do**
7:         **if** Human takes over **then**
8:             Human takes action $a_h \sim \pi_h(s_t)$.
9:             Add $(s_t, a_h)$ to the human buffer $\mathcal{D}_h$.
10:            Agent samples action $a_n \sim \pi_n(s_t)$.
11:            Predict future trajectory $\tau' = f(s_t, a_n, L) = (s_t, \tilde{s}_{t+1}, \cdots, \tilde{s}_{t+L})$.
12:            Add $(\tilde{s}, a_h, a_n)$ to the preference dataset $\mathcal{D}_{\text{pref}}$ for each $\tilde{s}$ in $(s_t, \tilde{s}_{t+1}, \cdots, \tilde{s}_{t+L})$.
13:            Observe $s_{t+1} \sim \mathcal{P}(\cdot \mid s_t, a_h)$.
14:         **else**
15:            Agent samples action $a_n \sim \pi_n(s_t)$.
16:            Observe $s_{t+1} \sim \mathcal{P}(\cdot \mid s_t, a_n)$.
17:         **end if**
18:         Train policy $\pi_n$ with loss function Eq. 4.
19:     **end for**
20: **end for**
21: **Output:** Policy $\pi_n$.

---

# B  Additional Experimental Results

We report the performance of our PPL and all the baselines with neural experts as proxy human policies in MetaDrive (Table 3), Table Wiping (Table 4), and Nut Assembly (Table 5) tasks, respectively. The test success rate curves of all three tasks are shown in Fig. 3.

Table 3: Comparison of methods with training/testing statistics in the MetaDrive environment with the neural expert as the proxy human policy. The overall intervention rate is given together with the expert data usage.

| Method | Expert-in-the-Loop | Training | | Testing | | |
|---|---|---|---|---|---|---|
| | | Expert Data Usage | Total Data Usage | Success Rate | Episodic Return | Route Completion |
| Neural Expert | – | – | – | $0.83_{\pm 0.07}$ | $340.2_{\pm 15.9}$ | $0.93_{\pm 0.02}$ |
| BC | ✗ | 20K | – | $0.12_{\pm 0.04}$ | $142.7_{\pm 27.5}$ | $0.46_{\pm 0.07}$ |
| GAIL | ✗ | 20K | 1M | $0.34_{\pm 0.08}$ | $196.5_{\pm 14.1}$ | $0.60_{\pm 0.09}$ |
| Ensemble-DAgger | ✓ | 3.2K (0.32) | 10K | $0.41_{\pm 0.08}$ | $238.6_{\pm 13.0}$ | $0.69_{\pm 0.07}$ |
| Thrifty-DAgger | ✓ | 2.9K (0.29) | 10K | $0.49_{\pm 0.07}$ | $248.2_{\pm 27.8}$ | $0.75_{\pm 0.06}$ |
| PVP | ✓ | 2.5K (0.25) | 10K | $0.56_{\pm 0.07}$ | $258.1_{\pm 23.4}$ | $0.76_{\pm 0.05}$ |
| IWR | ✓ | 2.7K (0.27) | 10K | $0.33_{\pm 0.11}$ | $217.0_{\pm 20.9}$ | $0.67_{\pm 0.06}$ |
| EIL | ✓ | 3.9K (0.39) | 10K | $0.11_{\pm 0.06}$ | $131.8_{\pm 29.5}$ | $0.42_{\pm 0.11}$ |
| HACO | ✓ | 2.6K (0.26) | 10K | $0.36_{\pm 0.15}$ | $210.2_{\pm 25.2}$ | $0.64_{\pm 0.10}$ |
| PPL (Ours) | ✓ | **1.2K** (0.20) | **6K** | $\mathbf{0.80}_{\pm 0.04}$ | $329.9_{\pm 13.4}$ | $\mathbf{0.92}_{\pm 0.03}$ |

Table 4: Results of different approaches in Table Wiping.

| Method | Expert Data Usage | Total Data | Success Rate |
|---|---|---|---|
| Neural Expert | – | – | 0.84 |
| BC | 10K | – | 0.11 |
| GAIL | 10K | 1M | 0.37 |
| PVP | 2.3K | 4K | 0.58 |
| IWR | 2.5K | 4K | 0.51 |
| EIL | 2.4K | 4K | 0.53 |
| HACO | 2.9K | 4K | 0.48 |
| PPL (Ours) | **0.2K** | **2K** | **0.80** |

Table 5: Results of different approaches in Nut Assembly.

| Method | Expert Data Usage | Total Data | Success Rate |
|---|---|---|---|
| Neural Expert | – | – | 0.60 |
| BC | 100K | – | 0.02 |
| GAIL | 100K | 1M | 0.08 |
| PVP | 49K | 200K | 0.35 |
| IWR | 54K | 200K | 0.29 |
| EIL | 48K | 200K | 0.25 |
| HACO | 77K | 200K | 0.15 |
| PPL (Ours) | **48K** | **200K** | **0.51** |

The neural experts in Table 3, 4, and 5 are trained with PPO-Lagrangian [33] for 20M environment steps, yet their test success rates still fall short of 100% for the following reasons. The MetaDrive safety environments occasionally generate rare but challenging scenarios that even a well-trained policy may fail to handle. In the Table Wiping task, the neural expert sometimes fails to remove one or two markings on the whiteboard, leaving a small patch of dirt uncleaned. In the Nut Assembly task, successful grasping requires the gripper to be precisely aligned with the metal ring's handle, which is highly sensitive to even minor action errors.

## C  Demo Video

We have attached our demo video to `https://metadriverse.github.io/ppl`. This video shows how we conduct human experiments and the evaluation results of our method Predictive Preference Learning from Human Interventions (PPL). This video includes five sections:

1. The first section gives an overview of Predictive Preference Learning, showing what human observes on the screen and how human provides corrective demonstrations in an episode.

2. The second section is the footage of the MetaDrive human experiment, where the human expert interacts with the driving agent via a gamepad.

3. The third section shows the evaluation results of the PPL agent in a held-out MetaDrive test environment. We compare our approach PPL with the PVP baseline [28], and both agents are trained to 10K timesteps. The evaluation results show that our approach PPL has a higher test success rate and lower safety cost.

4. The fourth section shows the applicability of our methods to manipulation tasks in Robosuite [49]: Table Wiping and Nut Assembly. PPL successfully imitates the expert and accomplishes both tasks in evaluation environments.

5. In the fifth section, we provide a full training session on MetaDrive. The video is played at 5× speed, and it shows how a human expert trains a PPL agent on MetaDrive in under 12 minutes, with approximately 1.8K demonstration steps and 10K environment steps.

## D  Human Subject Research Protocol

**Human Participants.**   Five university students (ages 20–30) with valid driver's licenses and video gaming experience took part in the study voluntarily. After receiving a detailed overview of the procedures and providing written informed consent under an IRB-approved protocol, each participant completed a hands-on familiarization session. During this session, they were informed how the predicted trajectories were shown on screen, and they practiced with our control interface and learning environments until performing ten consecutive successful runs before the main experiments.

**Main Experiment.**   Each participant began with one or two fully manual episodes to build confidence, and then ceded control to the agent when they felt safe. Participants were instructed to intervene only when the agent's predicted trajectory appeared unsafe, illegal, or inconsistent with their desired actions. They were directed to prioritize safe task completion and then to guide the agent toward their personal driving or manipulation preferences.

## E  Notations

Before we prove Theorem 4.1, we recall all the notations in this work. We denote the human policy as $\pi_h$ and the novice policy as $\pi_n$. For any stochastic policy $\pi(a \mid s)$ and the initial state distribution $d_0$ on state space $\mathcal{S}$, we define the value function $J(\pi)$ as the expected cumulative return: $J(\pi) = \mathbb{E}_{\tau \sim P_\pi} [\sum_{t=0}^{\infty} \gamma^t r(s_t, a_t)]$, wherein $\tau = (s_0, a_0, s_1, a_1, ...)$ is the trajectory sampled from trajectory distribution $P_\pi$ induced by $\pi$, $d_0$ and the state transition distribution $\mathcal{P}$. We also denote the Q-function of policy $\pi$ as $Q(s, a) = \mathbb{E}_{\tau \sim P_\pi} [\sum_{t=0}^{\infty} \gamma^t r(s_t, a_t) \big| s_0 = s, a_0 = a]$. And we define the discounted state distribution under $\pi$ as $d_\pi(s) = (1 - \gamma) \mathbb{E}_{\tau \sim P_\pi} [\sum_{t=0}^{\infty} \gamma^t \mathbb{I}[s_t = s]]$.

In our algorithm PPL, we have a preference dataset $\mathcal{D}_{\text{pref}}$ containing preference pairs $(s, a^+, a^-)$. The preference loss function of policy $\pi$ in PPL is defined as

$$\mathcal{L}_{\text{pref}}(\pi) = |\mathcal{D}_{\text{pref}}|^{-1} \sum_{(s', a^+, a^-) \in \mathcal{D}_{\text{pref}}} \left[ -\log \sigma \left( \beta \log \pi(a^+ \mid s) - \beta \log \pi(a^- \mid s) \right) \right], \qquad (7)$$

where $\beta$ is a positive constant and $\sigma(x) = (1 + \exp(-x))^{-1}$ is the Sigmoid function.

We denote the state distribution in $\mathcal{D}_{\text{pref}}$ as

$$d_{\text{pref}}(s) = |\mathcal{D}_{\text{pref}}|^{-1} \sum_{(s', a^+, a^-) \in \mathcal{D}_{\text{pref}}} \mathbb{I}[s' = s]. \qquad (8)$$

In addition, for any state $s$ in $\mathcal{D}_{\text{pref}}$ with $d_{\text{pref}}(s) > 0$, we denote the empirical preference-pair distribution in state $s$ as

$$\rho_{\text{pref}}^s(a_h, a_n) = \frac{\sum\limits_{(s', a^+, a^-) \in \mathcal{D}_{\text{pref}}} \mathbb{I}[s' = s, a^+ = a_h, a^- = a_n]}{\sum\limits_{(s', a^+, a^-) \in \mathcal{D}_{\text{pref}}} \mathbb{I}[s' = s]}, \qquad (9)$$

which is a distribution on $\mathcal{A} \times \mathcal{A}$.

## F Proof of Theorem 4.1

Our goal is to prove that the performance gap $J(\pi_h) - J(\pi_n)$ between the human policy $\pi_h$ and the agent policy $\pi_n$ can be bounded by the following three error terms: the state distribution shift $\delta_{\text{dist}}$, the quality of preference labels $\delta_{\text{pref}}$, and the optimization error $\epsilon$. We denote the total variation for any two distributions $P, Q$ on the same space as $D_{\text{TV}}(P, Q) = \frac{1}{2} \|P - Q\|_1$.

Here, we formally define the three error terms. The first state distribution shift error arises from the difference between the distribution of states in the preference dataset $\mathcal{D}_{\text{pref}}$ (denoted as $d_{\text{pref}}(s)$) and the discounted state distribution of the agent's policy $\pi_n$ (denoted as $d_{\pi_n}(s)$). To define the distribution shift error $\delta_{\text{dist}}$ in PPL, we use the total variation between the two distributions, i.e.,

$$\delta_{\text{dist}} = D_{\text{TV}}(d_{\pi_n}, d_{\text{pref}}). \qquad (10)$$

The second error term arises from the misalignment of the positive actions in the preference dataset, as the human action $a_h$ in each tuple $(\tilde{s}_i, a_h, a_n) \in \mathcal{D}_{\text{pref}}$ is sampled in state $s$ instead of the predicted future state $\tilde{s}_i$. In an ideal preference dataset, one would observe expert and novice actions drawn directly at $\tilde{s}_i$. To quantify this error, we define the following distribution $\rho_{\text{ideal}}^s(a_h, a_n) = \pi_h(a_h \mid s)\pi_n(a_n \mid s)$ on $\mathcal{A} \times \mathcal{A}$ for any state $s$, i.e., the distribution over pairs $(a_h, a_n)$ if both policies were sampled at directly at state $s$. Then we use

$$\delta_{\text{pref}} = \mathbb{E}_{s \sim d_{\text{pref}}} D_{\text{TV}}(\rho_{\text{ideal}}^s, \rho_{\text{pref}}^s) \qquad (11)$$

to define the errors in the preference dataset.

Finally, we define the optimization error of the agent policy $\pi_n$ as

$$\epsilon = \mathcal{L}_{\text{pref}}(\pi_n) - \mathcal{L}_{\text{pref}}(\pi_h). \qquad (12)$$

Under these notations, we have the following Thm. F.1. We note that when we choose a small $\beta \leq M^{-2}$ ($M$ is defined in Thm. F.1), we have

$$J(\pi_h) - J(\pi_n) = \frac{1}{1 - \gamma} \cdot O\left( \sqrt{\frac{\epsilon + 4 \log 2 \cdot \delta_{\text{pref}}}{2\beta}} + 2\delta_{\text{dist}} \right). \qquad (13)$$

**Theorem F.1** (Formal Statement of Theorem 4.1). *We denote the Q-function of the human policy $\pi_h$ as $Q^*(s, a)$. We assume that for any $(s, a, a')$, $|Q^*(s, a) - Q^*(s, a')| \leq U$, $|\log \pi_h(a|s) - \log \pi_h(a'|s)| \leq M$, and $|\log \pi_n(a|s) - \log \pi_n(a'|s)| \leq M$, where $U, M$ are positive constants. Then, we have*

$$J(\pi_h) - J(\pi_n) \leq \frac{U}{1 - \gamma} \cdot \left( \sqrt{\frac{\epsilon + 4(\beta M + \log 2) \cdot \delta_{\text{pref}}}{2\beta}} + \frac{\beta M^2}{8} + 2\delta_{\text{dist}} \right). \qquad (14)$$

*Proof.* The key is to combine Lem. F.2, Lem. F.3, and Lem. F.4 to obtain the bound.

From Lem. F.2, we can use the state distribution shift and the total variation of the two policies $\pi_h, \pi_n$ on $d_{\text{pref}}$ to bound the optimality gap:

$$J(\pi_h) - J(\pi_n) \leq \frac{U}{1-\gamma} \cdot \Big( \mathop{\mathbb{E}}_{s \sim d_{\text{pref}}} D_{\text{TV}}\big(\pi_h(\cdot|s), \pi_n(\cdot|s)\big) + 2\delta_{\text{dist}} \Big). \tag{15}$$

In addition, for any policy $\pi$, we define the function

$$g(\pi) = \mathop{\mathbb{E}}_{s \sim d_{\text{pref}}} \mathop{\mathbb{E}}_{a^+ \sim \pi_h(s), a^- \sim \pi_n(s)} \Big[ -\log \sigma \big( \beta \log \pi(a^+ \mid s) - \beta \log \pi(a^- \mid s) \big) \Big], \tag{16}$$

which represents the preference loss on ideal preference pairs, where $a^+, a^-$ are sampled directly at each state $s$.

Using F.4, we can bound the total variation term $\mathop{\mathbb{E}}_{s \sim d_{\text{pref}}} D_{\text{TV}}\big(\pi_h(\cdot|s)\big)$ by $g(\pi_n) - g(\pi_h)$:

$$\mathop{\mathbb{E}}_{s \sim d_{\text{pref}}} D_{\text{TV}}\big(\pi_h(\cdot|s), \pi_n(\cdot|s)\big) \leq \sqrt{\frac{g(\pi_n) - g(\pi_h)}{2\beta} + \frac{\beta M^2}{8}}. \tag{17}$$

In addition, by Lem. F.3, we can also bound $g(\pi_n) - g(\pi_h)$ by the optimization error $\epsilon$ on $\mathcal{L}_{\text{pref}}$ and the misalignment of preference levels $\delta_{\text{pref}}$:

$$g(\pi_n) - g(\pi_h) \leq \epsilon + 4(\beta M + \log 2) \cdot \delta_{\text{pref}}. \tag{18}$$

Combining Eq. 15, 17, and 18 yields Eq. 14.

$\square$

**Lemma F.2** (Performance Optimality Gap on the State Distribution Shift). *We recall that $d_{\text{pref}}(s) = |\mathcal{D}_{\text{pref}}|^{-1} \mathop{\mathbb{E}}_{(s', a^+, a^-) \sim \mathcal{D}_{\text{pref}}} \mathbb{I}[s' = s]$, and we define $U = \max_{s \in \mathcal{S}, a_1, a_2 \in \mathcal{A}} |Q^*(s, a_1) - Q^*(s, a_2)|$.*

*Then, for any two stochastic policies $\pi_h, \pi_n$, we have*

$$J(\pi_h) - J(\pi_n) \leq \frac{U}{1-\gamma} \cdot \Big( \mathop{\mathbb{E}}_{s \sim d_{\text{pref}}} D_{\text{TV}}\big(\pi_h(\cdot|s), \pi_n(\cdot|s)\big) + 2\delta_{\text{dist}} \Big). \tag{19}$$

*where $\delta_{\text{dist}} = D_{\text{TV}}(d_{\pi_n}, d_{\text{pref}})$.*

*Proof Sketch.* The key is to use the Performance Difference Lemma (Lem. G.2) on $J(\pi_h) - J(\pi_n)$, yielding Eq. 20. Then, we can apply Lem. G.1, which bounds the expectation on $s \sim d_{\text{pref}}$ and $s \sim d_{\pi_n}$ by the distribution shift term $\delta_{\text{dist}}$. Finally, applying the assumption $U = \max_{s \in \mathcal{S}, a_1, a_2 \in \mathcal{A}} |Q^*(s, a_1) - Q^*(s, a_2)|$ bounds the difference of the Q-function by the total variation between $\pi_h$ and $\pi_n$. $\square$

*Proof.* By the Performance Difference Lemma (Lem. G.2), we have

$$J(\pi_h) - J(\pi_n) = \frac{1}{1-\gamma} \mathop{\mathbb{E}}_{s \sim d_{\pi_n}} \mathop{\mathbb{E}}_{a_h \sim \pi_h(s), a_n \sim \pi_n(s)} [Q^*(s, a_h) - Q^*(s, a_n)]. \tag{20}$$

By Lem. G.1, as $d_{\pi_n}$ and $d_{\text{pref}}$ are two distributions on the same state space $\mathcal{S}$, we have

$$\begin{aligned}
&(1 - \gamma)(J(\pi_h) - J(\pi_n)) \\
&\leq \mathop{\mathbb{E}}_{s \sim d_{\text{pref}}} \mathop{\mathbb{E}}_{a_h \sim \pi_h(s), a_n \sim \pi_n(s)} [Q^*(s, a_h) - Q^*(s, a_n)] \\
&\quad + 2D_{\text{TV}}(d_{\pi_n}, d_{\text{pref}}) \cdot \max_{s \in \mathcal{S}} \left| \mathop{\mathbb{E}}_{a_h \sim \pi_h(s), a_n \sim \pi_n(s)} [Q^*(s, a_h) - Q^*(s, a_n)] \right| \\
&\leq \mathop{\mathbb{E}}_{s \sim d_{\text{pref}}} \mathop{\mathbb{E}}_{a_h \sim \pi_h(s), a_n \sim \pi_n(s)} [Q^*(s, a_h) - Q^*(s, a_n)] \\
&\quad + 2\delta_{\text{dist}} \cdot \max_{s \in \mathcal{S}} \mathop{\mathbb{E}}_{a_h \sim \pi_h(s), a_n \sim \pi_n(s)} |Q^*(s, a_h) - Q^*(s, a_n)| \\
&\leq \mathop{\mathbb{E}}_{s \sim d_{\text{pref}}} \left[ \mathop{\mathbb{E}}_{a_h \sim \pi_h(s)} Q^*(s, a_h) - \mathop{\mathbb{E}}_{a_n \sim \pi_n(s)} Q^*(s, a_n) \right] + 2U \cdot \delta_{\text{dist}},
\end{aligned} \tag{21}$$

where we use $U = \max\limits_{s \in \mathcal{S}, a_1, a_2 \in \mathcal{A}} |Q^*(s, a_1) - Q^*(s, a_2)|$ in the last inequality of Eq. 21.

In addition, $\pi_h(s)$ and $\pi_n(s)$ are two probability distributions on the same action space $\mathcal{A}$. By Lem. G.1, we have for any $s \in \mathcal{S}$,

$$\mathbb{E}_{a_h \sim \pi_h(s)} Q^*(s, a_h) - \mathbb{E}_{a_n \sim \pi_n(s)} Q^*(s, a_n) \le U \cdot D_{\mathrm{TV}}\big(\pi_h(\cdot|s), \pi_n(\cdot|s)\big). \tag{22}$$

This proves that

$$J(\pi_h) - J(\pi_n) \le \frac{U}{1 - \gamma} \cdot \Big( \mathbb{E}_{s \sim d_{\mathrm{pref}}} D_{\mathrm{TV}}\big(\pi_h(\cdot|s), \pi_n(\cdot|s)\big) + 2\delta_{\mathrm{dist}} \Big). \tag{23}$$

$\square$

**Lemma F.3** (Misalignment of Preference Pairs)**.** *We recall that the loss function of the policy $\pi$ is $\mathcal{L}_{pref}(\pi) = - \mathbb{E}_{(s, a^+, a^-) \sim \mathcal{D}_{pref}} [\log \sigma (\beta \log \pi(a^+ \mid s) - \beta \log \pi(a^- \mid s))]$. And the optimization loss is defined as $\epsilon = \mathcal{L}_{\mathrm{pref}}(\pi_n) - \mathcal{L}_{\mathrm{pref}}(\pi_h)$.*

*In addition, following Eq. 16, for any policy $\pi$, we define*

$$g(\pi) = \mathbb{E}_{s \sim d_{\mathrm{pref}}} \mathbb{E}_{a^+ \sim \pi_h(s), a^- \sim \pi_n(s)} \left[ -\log \sigma \big(\beta \log \pi(a^+ \mid s) - \beta \log \pi(a^- \mid s)\big) \right]. \tag{24}$$

*Under the assumption that for any $(s, a, a')$, $|\log \pi_h(a|s) - \log \pi_h(a'|s)| \le M$, and $|\log \pi_n(a|s) - \log \pi_n(a'|s)| \le M$, we have*

*we can bound*

$$g(\pi_n) - g(\pi_h) \le \epsilon + 4(\beta M + \log 2) \cdot \delta_{\mathrm{pref}}, \tag{25}$$

*where $\delta_{\mathrm{pref}} = \mathbb{E}_{s \sim d_{\mathrm{pref}}} D_{\mathrm{TV}}(\rho^s_{\mathrm{ideal}}, \rho^s_{\mathrm{pref}})$.*

*Proof Sketch.* The key is to apply Lem. G.1 on the two distributions $\rho^s_{\mathrm{ideal}}$ and $\rho^s_{\mathrm{pref}}$, so that we can bound the difference of $\mathcal{L}_{\mathrm{pref}}(\pi)$ and $g(\pi)$ for any policy $\pi$ by $O(\delta_{\mathrm{pref}})$. $\square$

*Proof.* For any $s \in \mathcal{S}$, we denote $\rho^s_{\mathrm{ideal}}(a_h, a_n) = \pi_h(a_h \mid s)\pi_n(a_n \mid s)$, a probability distribution on $\mathcal{A} \times \mathcal{A}$. We also denote $\rho^s_{\mathrm{pref}}(a_h, a_n) = \rho_{\mathrm{pref}}(s, a_h, a_n)/d_{\mathrm{pref}}(s)$ for any $s$ such that $d_{\mathrm{pref}}(s) > 0$, where we recall that $\rho_{\mathrm{pref}}(s, a_h, a_n) = |\mathcal{D}_{\mathrm{pref}}|^{-1} \mathbb{E}_{(s', a^+, a^-) \sim \mathcal{D}_{\mathrm{pref}}} \mathbb{I}[s' = s, a^+ = a_h, a^- = a_n]$.

The key is that $\rho^s_{\mathrm{ideal}}$ and $\rho^s_{\mathrm{pref}}$ are two distributions on the same space $\mathcal{A} \times \mathcal{A}$, and we can apply Lem. G.1 on Eq. 24 to obtain the proof.

We denote $l^\pi(s, a^+, a^-) = -\log \sigma (\beta \log \pi(a^+ \mid s) - \beta \log \pi(a^- \mid s))$.

We also denote $l^\pi_{\max} = \max\limits_{s, a^+, a^-} |l^\pi(s, a^+, a^-)|$, and $l_{\max} = \max(l^{\pi_h}_{\max}, l^{\pi_n}_{\max})$. Then, for any policy $\pi$,

$$\begin{aligned}
g(\pi) &= \mathbb{E}_{s \sim d_{\mathrm{pref}}} \mathbb{E}_{a^+ \sim \pi_h(s), a^- \sim \pi_n(s)} l^\pi(s, a^+, a^-) \\
&= \mathbb{E}_{s \sim d_{\mathrm{pref}}} \mathbb{E}_{(a^+, a^-) \sim \rho^s_{\mathrm{ideal}}} l^\pi(s, a^+, a^-) \\
&\le \mathbb{E}_{s \sim d_{\mathrm{pref}}} \left[ 2l^\pi_{\max} \cdot D_{\mathrm{TV}}(\rho^s_{\mathrm{ideal}}, \rho^s_{\mathrm{pref}}) + \mathbb{E}_{(a^+, a^-) \sim \rho^s_{\mathrm{pref}}} l^\pi(s, a^+, a^-) \right] \\
&= 2l^\pi_{\max}\delta_{\mathrm{pref}} + \mathbb{E}_{s \sim d_{\mathrm{pref}}} \mathbb{E}_{(a^+, a^-) \sim \rho^s_{\mathrm{pref}}} l^\pi(s, a^+, a^-) \\
&= 2l^\pi_{\max}\delta_{\mathrm{pref}} + |\mathcal{D}_{\mathrm{pref}}|^{-1} \sum_{(s, a^+, a^-) \in \mathcal{D}_{\mathrm{pref}}} l^\pi(s, a^+, a^-) \\
&= 2l^\pi_{\max}\delta_{\mathrm{pref}} + \mathcal{L}_{\mathrm{pref}}(\pi).
\end{aligned} \tag{26}$$

Similarly, we can obtain that $g(\pi) \ge -2l^\pi_{\max}\delta_{\mathrm{pref}} + \mathcal{L}_{\mathrm{pref}}(\pi)$ for any policy $\pi$. Thus we have

$$g(\pi_n) - g(\pi_h) \le 4l_{\max}\delta_{\mathrm{pref}} + (\mathcal{L}_{\mathrm{pref}}(\pi_n) - \mathcal{L}_{\mathrm{pref}}(\pi_h)) = 4l_{\max}\delta_{\mathrm{pref}} + \epsilon. \tag{27}$$

Finally, under the condition that $|\log \pi(a|s) - \log \pi(a'|s)| \leq M$ for any $(s, a, a')$, we have $|l^\pi(s, a, a')| \leq -\log \sigma(-\beta M) = \log(1 + \exp(\beta M)) \leq \beta M + \log 2$.

This implies that $l_{\max} \leq \beta M + \log 2$ and completes the proof. $\qquad\square$

**Lemma F.4** (Optimization Error Bounds the Total Variation). *We assume that for any $(s, a, a')$, $|\log \pi_h(a|s) - \log \pi_h(a'|s)| \leq M$, and $|\log \pi_n(a|s) - \log \pi_n(a'|s)| \leq M$.*

*We recall that $g(\pi) = \mathbb{E}_{s \sim d_{\mathrm{pref}}} \mathbb{E}_{a^+ \sim \pi_h(s), a^- \sim \pi_n(s)} [-\log \sigma (\beta \log \pi(a^+ \mid s) - \beta \log \pi(a^- \mid s))]$, which is defined in Eq. 16. Then we have*

$$\mathbb{E}_{s \sim d_{\mathrm{pref}}} D_{\mathrm{TV}}\big(\pi_h(\cdot|s), \pi_n(\cdot|s)\big) \leq \sqrt{\frac{g(\pi_n) - g(\pi_h)}{2\beta} + \frac{\beta M^2}{8}}. \tag{28}$$

*Proof Sketch.* First, we define

$$f(\pi) = -\frac{\beta}{2} \mathbb{E}_{s \sim d_{\mathrm{pref}}} \mathbb{E}_{a^+ \sim \pi_h(s), a^- \sim \pi_n(s)} \left[ \log \pi(a^+|s) - \log \pi(a^-|s) \right] + \log 2. \tag{29}$$

Using the Taylor's expansion on the function $\log \sigma(x)$ at $x = 0$, when the policy $\pi$ satisfies $|\log \pi(a|s) - \log \pi(a'|s)| \leq M$ for any $(s, a, a')$, we can obtain that $|g(\pi) - f(\pi)| \leq \frac{\beta^2 M^2}{8}$.

In addition, $f(\pi_n) - f(\pi_h)$ bounds the KL divergence of the two policies $\pi_h$ and $\pi_n$ over $s \sim d_{\mathrm{pref}}$. So we can use Pinsker's inequality to obtain the bound on $D_{\mathrm{TV}}\big(\pi_h(\cdot|s), \pi_n(\cdot|s)\big)$. $\qquad\square$

*Proof.* For any $(s, a^+, a^-)$, we denote $u_n(s, a^+, a^-) = \log \pi_n(a^+|s) - \log \pi_n(a^-|s)$, and $u_h(s, a^+, a^-) = \log \pi_h(a^+|s) - \log \pi_h(a^-|s)$. From the assumptions, we can obtain that $|u_n(s, a^+, a^-)| \leq M$ and $|u_h(s, a^+, a^-)| \leq M$.

By definition of the function $g(\cdot)$, we have $g(\pi_n) - g(\pi_h) = \mathbb{E}_{s \sim d_{\mathrm{pref}}} \mathbb{E}_{a^+ \sim \pi_h(s), a^- \sim \pi_n(s)} [\log \sigma (\beta \cdot u_h(s, a^+, a^-)) - \log \sigma (\beta \cdot u_n(s, a^+, a^-))]$.

The Taylor's expansion of $\log \sigma(x)$ at $x = 0$ ensures that for any $x \in \mathbb{R}$, we have

$$\left| \log \sigma(x) + \log 2 - \frac{1}{2}x \right| \leq \frac{1}{8}x^2. \tag{30}$$

This ensures that

$$g(\pi_n) = \mathbb{E}_{s \sim d_{\mathrm{pref}}} \mathbb{E}_{a^+ \sim \pi_h(s), a^- \sim \pi_n(s)} \left[ -\log \sigma \big(\beta \cdot u_n(s, a^+, a^-)\big) \right]$$
$$\geq \log 2 - \frac{\beta}{2} \mathbb{E}_{s \sim d_{\mathrm{pref}}} \mathbb{E}_{a^+ \sim \pi_h(s), a^- \sim \pi_n(s)} u_n(s, a^+, a^-) - \frac{\beta^2 M^2}{8}, \tag{31}$$

and similarly,

$$g(\pi_h) = \mathbb{E}_{s \sim d_{\mathrm{pref}}} \mathbb{E}_{a^+ \sim \pi_h(s), a^- \sim \pi_n(s)} \left[ -\log \sigma \big(\beta \cdot u_h(s, a^+, a^-)\big) \right]$$
$$\leq \log 2 - \frac{\beta}{2} \mathbb{E}_{s \sim d_{\mathrm{pref}}} \mathbb{E}_{a^+ \sim \pi_h(s), a^- \sim \pi_n(s)} u_h(s, a^+, a^-) + \frac{\beta^2 M^2}{8}, \tag{32}$$

Hence, we have

$$g(\pi_n) - g(\pi_h) \geq \frac{\beta}{2} \mathbb{E}_{s \sim d_{\mathrm{pref}}} \mathbb{E}_{a^+ \sim \pi_h(s), a^- \sim \pi_n(s)} \left[ u_h(s, a^+, a^-) - u_n(s, a^+, a^-) \right] - \frac{\beta^2 M^2}{4}$$
$$= \frac{\beta}{2} \mathbb{E}_{s \sim d_{\mathrm{pref}}} \mathbb{E}_{a^+ \sim \pi_h(s), a^- \sim \pi_n(s)} \left[ \log \frac{\pi_h(a^+|s)}{\pi_h(a^-|s)} - \log \frac{\pi_n(a^+|s)}{\pi_n(a^-|s)} \right] - \frac{\beta^2 M^2}{4} \tag{33}$$
$$= \frac{\beta}{2} \mathbb{E}_{s \sim d_{\mathrm{pref}}} \left[ \mathbb{E}_{a^+ \sim \pi_h(s)} \log \frac{\pi_h(a^+|s)}{\pi_n(a^+|s)} + \mathbb{E}_{a^- \sim \pi_n(s)} \log \frac{\pi_n(a^-|s)}{\pi_h(a^-|s)} \right] - \frac{\beta^2 M^2}{4}.$$

By the definition of KL divergence, we have

$$g(\pi_n) - g(\pi_h) = \frac{\beta}{2} \mathop{\mathbb{E}}_{s \sim d_{\mathrm{pref}}} \left[ \mathrm{KL}\Big(\pi_h(\cdot|s)\big\|\pi_n(\cdot|s)\Big) + \mathrm{KL}\Big(\pi_n(\cdot|s)\big\|\pi_h(\cdot|s)\Big) \right] - \frac{\beta^2 M^2}{4}$$

$$\geq 2\beta \mathop{\mathbb{E}}_{s \sim d_{\mathrm{pref}}} \left[ D_{\mathrm{TV}}\big(\pi_h(\cdot|s), \pi_n(\cdot|s)\big) \right]^2 - \frac{\beta^2 M^2}{4}, \tag{34}$$

where we use Pinsker's inequality to obtain the bound on $D_{\mathrm{TV}}\big(\pi_h(\cdot|s), \pi_n(\cdot|s)\big)$ from the KL divergence.

Finally, we apply the inequality $\mathbb{E}[X^2] \geq (\mathbb{E}[X])^2$ on $X = D_{\mathrm{TV}}\big(\pi_h(\cdot|s), \pi_n(\cdot|s)\big)$, so that we have

$$g(\pi_n) - g(\pi_h) \geq 2\beta \left[ \mathop{\mathbb{E}}_{s \sim d_{\mathrm{pref}}} D_{\mathrm{TV}}\big(\pi_h(\cdot|s), \pi_n(\cdot|s)\big) \right]^2 - \frac{\beta^2 M^2}{4}. \tag{35}$$

This proves that

$$\mathop{\mathbb{E}}_{s \sim d_{\mathrm{pref}}} D_{\mathrm{TV}}\big(\pi_h(\cdot|s), \pi_n(\cdot|s)\big) \leq \sqrt{\frac{g(\pi_n) - g(\pi_h)}{2\beta} + \frac{\beta M^2}{8}}. \tag{36}$$

$\square$

## G  Technical Lemmas

**Lemma G.1** (Expectation Difference via Total Variation). *Let $P$ and $Q$ be two probability distributions on a measurable space $\mathcal{X}$, and let $f : \mathcal{X} \to \mathbb{R}$ be any measurable function satisfying the uniform bound $|f(x)| \leq M$ for any $x \in \mathcal{X}$. Then*

$$\left| \mathop{\mathbb{E}}_{x \sim P(\cdot)} f(x) - \mathop{\mathbb{E}}_{x \sim Q(\cdot)} f(x) \right| \leq 2M \cdot D_{\mathrm{TV}}(P, Q), \tag{37}$$

*where $D_{\mathrm{TV}}(P, Q) = \frac{1}{2}\|P - Q\|_1$ is the total variation distance.*

*In addition, when the measurable function $g$ satisfies the bound $|g(x_1) - g(x_2)| \leq M'$ for any $x_1, x_2 \in \mathcal{X}$, we have*

$$\left| \mathop{\mathbb{E}}_{x \sim P(\cdot)} g(x) - \mathop{\mathbb{E}}_{x \sim Q(\cdot)} g(x) \right| \leq M' \cdot D_{\mathrm{TV}}(P, Q). \tag{38}$$

*Proof.* When $|f(x)| \leq M$ for any $x$, we have

$$\left| \mathop{\mathbb{E}}_{x \sim P(\cdot)} f(x) - \mathop{\mathbb{E}}_{x \sim Q(\cdot)} f(x) \right| = \left| \sum_x f(x) \cdot (P(x) - Q(x)) \right|$$

$$\leq \sum_x |f(x)| \cdot |P(x) - Q(x)|$$

$$\leq M \cdot \sum_x |P(x) - Q(x)| \tag{39}$$

$$= 2M \cdot D_{\mathrm{TV}}(P, Q).$$

When $|g(x_1) - g(x_2)| \leq M'$ for any $x_1, x_2 \in \mathcal{X}$, we set $f(x) = g(x) - \frac{1}{2}c$, where $c = \sup_{x \in \mathcal{X}} f(x) + \inf_{x \in \mathcal{X}} f(x)$. As we have $|f(x)| \leq M'/2$ for any $x \in \mathcal{X}$, we have

$$\left| \mathop{\mathbb{E}}_{x \sim P(\cdot)} g(x) - \mathop{\mathbb{E}}_{x \sim Q(\cdot)} g(x) \right| = \left| \mathop{\mathbb{E}}_{x \sim P(\cdot)} f(x) - \mathop{\mathbb{E}}_{x \sim Q(\cdot)} f(x) \right| \leq M' \cdot D_{\mathrm{TV}}(P, Q). \tag{40}$$

$\square$

**Lemma G.2** (Performance Gap Between Human Policy and Novice Policy). *We denote the Q-function of human policy $\pi_h$ as $Q^*(s, a) = \mathop{\mathbb{E}}\limits_{\tau \sim P_{\pi_h}} [\sum\limits_{t=0}^{\infty} \gamma^t r(s_t, a_t) | s_0 = s, a_0 = a].$*

*For the human policy $\pi_h$ and the novice policy $\pi_n$ whose value functions are $J(\pi_h), J(\pi_n)$, respectively, we have*

$$J(\pi_h) - J(\pi_n) = \frac{1}{1 - \gamma} \mathop{\mathbb{E}}\limits_{s \sim d_{\pi_n}} \left[ \mathop{\mathbb{E}}\limits_{a_h \sim \pi_h(s)} Q^*(s, a_h) - \mathop{\mathbb{E}}\limits_{a_n \sim \pi_n(s)} Q^*(s, a_n) \right]. \tag{41}$$

*Proof.* We denote the Q-function of novice policy $\pi_n$ as $Q_n(s, a) = \mathop{\mathbb{E}}\limits_{\tau \sim P_{\pi_n}} [\sum\limits_{t=0}^{\infty} \gamma^t r(s_t, a_t) | s_0 = s, a_0 = a].$

We denote value functions of $\pi_h, \pi_n$ as $V^*(s) = \mathop{\mathbb{E}}\limits_{a \sim \pi_h(s)} Q^*(s, a)$ and $V_n(s) = \mathop{\mathbb{E}}\limits_{a \sim \pi_n(s)} Q_n(s, a)$, respectively. And we have $J(\pi_h) = \mathop{\mathbb{E}}\limits_{s_0 \sim d_0} V^*(s_0)$, and $J(\pi_n) = \mathop{\mathbb{E}}\limits_{s_0 \sim d_0} V_n(s_0)$.

We define the advantage function of $\pi_h$ as $A^*(s, a) = Q^*(s, a) - V^*(s)$.

By the performance difference lemma (Lemma 6.1, [14]), we have

$$\mathop{\mathbb{E}}\limits_{s_0 \sim d_0} \left[ V_n(s_0) - V^*(s_0) \right] = \frac{1}{1 - \gamma} \mathop{\mathbb{E}}\limits_{s \sim d_{\pi_n}} \left[ \mathop{\mathbb{E}}\limits_{a \sim \pi_n(s)} A^*(s, a) \right]. \tag{42}$$

This implies that

$$\mathop{\mathbb{E}}\limits_{s_0 \sim d_0} \left[ V_n(s_0) - V^*(s_0) \right] = \frac{1}{1 - \gamma} \mathop{\mathbb{E}}\limits_{s \sim d_{\pi_n}} \left[ \mathop{\mathbb{E}}\limits_{a \sim \pi_n(s)} [Q^*(s, a) - V^*(s)] \right]$$

$$= \frac{1}{1 - \gamma} \mathop{\mathbb{E}}\limits_{s \sim d_{\pi_n}} \left[ -V^*(s) + \mathop{\mathbb{E}}\limits_{a \sim \pi_n(s)} Q^*(s, a) \right] \tag{43}$$

$$= \frac{1}{1 - \gamma} \mathop{\mathbb{E}}\limits_{s \sim d_{\pi_n}} \left[ -\mathop{\mathbb{E}}\limits_{a \sim \pi_h(s)} Q^*(s, a) + \mathop{\mathbb{E}}\limits_{a \sim \pi_n(s)} Q^*(s, a) \right].$$

Multiplying $-1$ on both sides, we can obtain that

$$J(\pi_h) - J(\pi_n) = \frac{1}{1 - \gamma} \mathop{\mathbb{E}}\limits_{s \sim d_{\pi_n}} \left[ \mathop{\mathbb{E}}\limits_{a_h \sim \pi_h(s)} Q^*(s, a_h) - \mathop{\mathbb{E}}\limits_{a_n \sim \pi_n(s)} Q^*(s, a_n) \right]. \tag{44}$$

$\square$

# H   Ethics Statement

Our Predictive Preference Learning from Human Interventions (PPL) delivers a human-friendly, human-in-the-loop training process that increases automation while minimizing expert effort, advancing more intelligent AI systems with reduced human burden. All the experiments are conducted entirely in simulation, ensuring no physical risk to participants. All volunteers provided informed consent, were compensated above local market rates, and could pause or withdraw from the study at any time without penalty. Individual sessions lasted less than one hour, with a mandatory rest period of at least three hours before any subsequent participation. No personal or sensitive data was collected or shared. We have obtained the IRB approval to conduct this project.

While PPL promises positive social impact by streamlining human-AI collaboration, it may also encourage overreliance on automated systems or inherit biases present in expert involvement.

