# OpenReview forum: "Predictive Preference Learning from Human Interventions"
_NeurIPS.cc/2025/Conference — NeurIPS 2025 spotlight_

### Official Review · Reviewer_p5vD · 2025-07-02

**Clarity:** 3
**Significance:** 2
**Originality:** 3
**Rating:** 4
**Confidence:** 3

**Summary:**

This paper proposes Predictive Preference Learning (PPL), an interactive imitation learning framework that improves learning efficiency by converting proactive human interventions into contrastive preference signals. Unlike traditional imitation learning, which reacts to past errors, PPL anticipates future agent behavior using a trajectory prediction model, allowing humans to intervene before failures occur. These interventions generate preference labels over predicted future states, enabling the agent to learn not only from direct demonstrations but also from ranked alternatives. The learning objective combines behavioral cloning with a contrastive preference optimization loss, encouraging the agent to prefer safer trajectories.

**Questions:**

Theoretical analysis in Theorem 4.1 states that $\beta$ must be “sufficiently small” to bound the performance gap, specifically assuming $\beta \leq M^{-2}$. In practice, would it be beneficial to conduct additional experiments varying both $\beta$ and $M$ to empirically validate this theoretical condition?

**Ethical Concerns:**

["NO or VERY MINOR ethics concerns only"]

**Final Justification:**

My concerns have been addressed.

**Limitations:**

See weaknesses and questions.

**Quality:**

3

**Strengths And Weaknesses:**

Strengths:
1. PPL leverages predicted future trajectories to convert human interventions into contrastive preference labels over forecasted states, enabling proactive error prevention rather than reactive correction. This addresses the distributional shift in traditional imitation learning and reduces the cognitive burden on human supervisors by visualizing potential failure scenarios.
2. Experiments on autonomous driving (MetaDrive) and robotic manipulation (Robosuite) tasks demonstrate that PPL achieves near-optimal policies with significantly fewer human demonstrations (e.g., 2.9K vs. up to 6.9K in baselines) and higher success rates (76% in MetaDrive with real humans), outperforming state-of-the-art methods like PVP and Ensemble-DAgger.

Weaknesses:
1. The paper acknowledges that all experiments are conducted in simulation (MetaDrive and Robosuite), lacking validation on physical robots. Simulated environments omit real-world challenges like sensor noise, mechanical imperfections, and unmodeled dynamics (e.g., gripper precision in Nut Assembly or traffic unpredictability in driving). A rigorous study on real robots is essential to verify PPL’s robustness.

---

> ### Author Rebuttal · Authors · 2025-07-31
>
> Thank you for your careful reading and constructive feedback on our manuscript. We have organized our replies to each of your questions below.
>
> __Strengths And Weaknesses:__
>
> >1. The paper acknowledges that all experiments are conducted in simulation (MetaDrive and Robosuite), lacking validation on physical robots.
>
> We appreciate the reviewer’s suggestion regarding real-world experimentation. However, we would like to emphasize that this work focuses on the algorithmic design and theoretical foundations of a new human-in-the-loop machine learning method. As a NeurIPS submission, our primary contribution lies in advancing the core machine learning methodology, which we rigorously validate in simulation environments with real human feedback. We agree that deploying the algorithm in physical systems is an important direction, and we plan to pursue real-world robotic experiments as part of future work.
>
> Moreover, __our current evaluations involve real human participants__, and Table 1 in our paper demonstrates that __PPL is a more human‑friendly and efficient interactive imitation learning method__. Table 1 shows that PPL yields a higher success rate (PPL 0.76 vs. 0.46 for the best baseline PVP) and reduces expert takeovers (PPL 2.9 K vs. PVP 4.9 K). The experiments with real human participants can help bridge the real-world robotic experiments in our future work.
>
> >  Simulated environments omit real-world challenges, so it is essential to verify PPL’s robustness.
>
> We perform additional experiments to show that __PPL is robust to both the noise in the trajectory prediction model and the perturbation in the expert actions__.
>
> First, we conducted additional experiments by injecting random noise into our trajectory prediction models. __With an imperfect predictive model, PPL still outperforms all the baselines.__ We inject random noise $e_{noise}$ to the outputs $s’$ of the trajectory predictor, and we set the norm $||e_{noise}||_2 = \epsilon * ||s’||_2$. Then we set the constant $\epsilon = 0, 0.1, 0.2, …$ to test PPL’s robustness to noise in trajectory predictors.
>
> Using the same setups as Tables 3–5 (MetaDrive, Table Wiping, and Nut Assembly), we find that with a noisy predictive model, PPL still outperforms all the baselines. In MetaDrive and Table Wiping, PPL keeps the best performance when the noise $\epsilon \leq 0.2$. In Nut Assembly, PPL outperforms all the baselines when $\epsilon \leq 0.1$.
>
> In MetaDrive, we report the result of PPL with 10K total training data.
>
> |MetaDrive |PPL ($\epsilon = 0$) | PPL ($\epsilon = 0.1$) | PPL ($\epsilon = 0.2$) |  PPL ($\epsilon = 0.3$) | PVP (best baseline) |
> |---|---|---|---|---|---|
> |Test Success Rate| 0.80 | 0.73 | 0.64 | 0.51 | 0.56 |
>
> In Table Wiping, we report the result of PPL and baselines with 2K total training data.
>
> |Table Wiping |PPL ($\epsilon = 0$) | PPL ($\epsilon = 0.1$) | PPL ($\epsilon = 0.2$) |  PPL ($\epsilon = 0.3$) | PVP (best baseline) |
> |---|---|---|---|---|---|
> |Test Success Rate| 0.80 | 0.76 | 0.70 | 0.56 | 0.58 |
>
> In Nut Assembly, we report the result of PPL and baselines with 200K total training data.
>
> | Nut Assembly|PPL ($\epsilon = 0$) | PPL ($\epsilon = 0.05$) | PPL ($\epsilon = 0.1$) |  PPL ($\epsilon = 0.15$) | PVP (best baseline) |
> |---|---|---|---|---|---|
> |Test Success Rate| 0.51 | 0.47 | 0.41 | 0.30 | 0.35 |
>
> Then, we show that __PPL is robust to noise and perturbation in the expert__. Following Table 3, we use a well-trained PPO expert as a proxy human policy, which achieves a 0.83 success rate. Then, we inject Gaussian noises $\mathcal{N}(0, \sigma^2 I)$ to the expert actions, and we evaluate how $\sigma$ affects the test success rate of PPL in MetaDrive with 10K total training steps. The result shows that PPL is robust to a noise $\sigma \leq 0.2$ in expert actions in MetaDrive.
>
> | Noise $\sigma$ in expert policy | Test Success Rate of PPL |
> |--------|--------|
> | 0    | 0.80  |
> | 0.1  | 0.77  |
> | 0.2  | 0.70  |
> | 0.3  | 0.62  |
> | 0.4  | 0.43  |
>
> __Questions:__
>
> >1.  Would it be beneficial to conduct additional experiments varying both beta and M to empirically validate this theoretical condition of $\beta \leq M^{-2}$?
>
> __Our experiment that varies $\beta$ in MetaDrive empirically validates this theoretical condition.__
>
> According to line 203, $M = \sup_{s, a, a’} |\log \pi_h(a|s) - \log \pi_h(a|s’) |$, which is a constant determined by the human policy over the entire state-action space and thus difficult to measure directly. To empirically probe the theoretical condition $\beta \leq M^{-2}$, we conduct an additional experiment that varies $\beta$ from 0.01 to 1.0 in MetaDrive, using a fixed budget of 10K training steps:
>
> | PPL | $\beta = 0.01$ | $\beta = 0.05$ | $\beta = 0.1$ | $\beta = 0.2$ | $\beta = 0.4$ |  $\beta = 0.6$ |   $\beta = 0.8$ | $\beta = 1.0$ |
> |---|---|---|---|---|---|---|---|---|
> |Test Success Rate| 0.61 | 0.73 | __0.80__ | __0.80__ | 0.77 | 0.69 | 0.58 | 0.47 |
>
> This empirically shows that the PPL performance degrades when $\beta$ is too large, due to the violation of $\beta \leq M^{-2}$. In addition, when $\beta$ is too small, the behavioral cloning loss dominates the preference loss $\mathcal{L}_\text{pref}$, harming PPL’s performance empirically.

---

> > ### Comment · Reviewer_p5vD · 2025-08-07
> >
> > Thanks for your efforts. Most of my concerns have been addressed.

---

### Official Review · Reviewer_RuYs · 2025-07-02

**Clarity:** 3
**Significance:** 3
**Originality:** 2
**Rating:** 4
**Confidence:** 4

**Summary:**

Predictive Preference Learning from Human Interventions (PPL) is an interactive imitation learning approach that utilizes predicted future trajectory and, upon expert override, converts the corrective action into contrastive preference labels over those predicted states. By propagating feedback across multiple future states rather than only the intervention point, PPL reduces the number of required expert interventions and demonstrations, boosting sample efficiency. Experimental results on MetaDrive and RoboSuite show that PPL outperforms baseline imitation learning methods. A theoretical analysis further demonstrates that choosing the optimal preference horizon L balances distribution shift against label quality.

**Questions:**

- **Choice of Horizon $L$:** How did you select the preference horizon $L$ for each benchmark? Is there an intuition or heuristic for choosing $L$ when applying PPL to new tasks?
- **Predictive Model Details:** Which trajectory prediction model was employed for RoboSuite experiments, and how sensitive is PPL to its accuracy?
- **Data‐Budget Fairness:** Why do different methods use varying amounts of expert data? Could you compare all methods using the same data budget?
- **Intervention Timing:** For each baseline and PPL, at what points during training or rollout are expert interventions incorporated into the learning process?

**Ethical Concerns:**

["NO or VERY MINOR ethics concerns only"]

**Final Justification:**

Additional experiments, including those with a noised transition model and tested for robustness to $L$, as well as comparisons with other baselines, effectively address my concerns.

**Limitations:**

yes

**Paper Formatting Concerns:**

No concern about paper formatting.

**Quality:**

3

**Strengths And Weaknesses:**

### Strengths

- **Intuitive Intervention‐to‐Preference Conversion**: Converting human overrides into contrastive preference labels over predicted trajectories is both simple and conceptually clear.
- **Cross‐Domain Performance**: Demonstrates superior sample efficiency and task performance across diverse environments, including AD and robotic manipulations.

### Weaknesses
- **Dependence on Accurate Predictive Models**: Requires a high‐fidelity trajectory predictor (e.g., a rule‐based physics model); applicability may be limited in domains without such models (e.g., precise robotic manipulation, reasoning in LLMs, etc.).
- **Theoretical Bounds Lack Practical Guidance**: While the analysis shows that the error varies with the preference horizon $L$, it does not offer intuition or heuristics for selecting $L$ in new domains, nor does it provide a proof of advantage over other methods.
- **Missing Comparisons to Key Interactive IL Work**: Omits citation and empirical comparison to recent human‐in‐the‐loop imitation‐learning methods [1], [2].

**References**\
[1] Liu et al., Robot Learning on the Job: Human‐in‐the‐Loop Autonomy and Learning During Deployment, IJRR 2024\
[2] Liu et al., Multi‐Task Interactive Robot Fleet Learning with Visual World Models, CoRL 2024

---

> ### Author Rebuttal · Authors · 2025-07-31
>
> We appreciate the time and effort you devoted to evaluating our work and offering valuable suggestions. We summarize and respond to each question as follows:
>
> __Strengths And Weaknesses:__
>
> >1. Dependence on Accurate Predictive Models: PPL requires a high‐fidelity trajectory predictor.
>
> First, we emphasize that __our trajectory predictor remains lightweight yet effective__. In our main experiment (Table 1), we use an L‑step simulator rollout that runs at up to 1,000 fps on a CPU. We also developed a faster trajectory extrapolation method by simulating a kinematic bicycle model via forward integration (See Lines 320–323), which achieves 3,000 fps. As reported in Table 2, this extrapolation approach attains a 0.78 test success rate in MetaDrive, nearly matching the 0.80 achieved by the rollout predictor. This demonstrates that in domains where a rollout-based predictor is not available, __our trajectory extrapolation method can serve as a replacement without impacting PPL's performance__.
>
> In addition, we perform additional experiments to show that PPL is robust to noises in trajectory predictors. __With an imperfect predictive model, PPL still outperforms all the baselines.__
>
> We inject random noise $e_{noise}$ to the outputs $s’$ of the trajectory predictor, and we set the norm $||e_{noise}||_2 = \epsilon * ||s’||_2$. Then we set the constant $\epsilon = 0, 0.1, 0.2, …$ to test PPL’s robustness to noises in trajectory predictors.
> We use MetaDrive, Table Wiping, and Nut Assembly environments following the same setup in Tables 3,4,5 from our paper, respectively. We find that with a noisy predictive model, PPL still outperforms all the baselines in MetaDrive and Table Wiping when the noise $\epsilon \leq 0.2$. In Nut Assembly, PPL outperforms the baselines when $\epsilon \leq 0.1$.
>
> In MetaDrive, we report the result of PPL with 10K total training data.
>
> |MetaDrive |PPL ($\epsilon = 0$) | PPL ($\epsilon = 0.1$) | PPL ($\epsilon = 0.2$) |  PPL ($\epsilon = 0.3$) | PVP (best baseline) |
> |---|---|---|---|---|---|
> |Test Success Rate| 0.80 | 0.73 | 0.64 | 0.51 | 0.56 |
>
> In Table Wiping, we report the result of PPL and baselines with 2K total training data.
>
> |Table Wiping |PPL ($\epsilon = 0$) | PPL ($\epsilon = 0.1$) | PPL ($\epsilon = 0.2$) |  PPL ($\epsilon = 0.3$) | PVP (best baseline) |
> |---|---|---|---|---|---|
> |Test Success Rate| 0.80 | 0.76 | 0.70 | 0.56 | 0.58 |
>
> In Nut Assembly, we report the result of PPL and baselines with 200K total training data.
>
>
> |Nut Assembly |PPL ($\epsilon = 0$) | PPL ($\epsilon = 0.05$) | PPL ($\epsilon = 0.1$) |  PPL ($\epsilon = 0.15$) | PVP (best baseline) |
> |---|---|---|---|---|---|
> |Test Success Rate| 0.51 | 0.47 | 0.41 | 0.30 | 0.35 |
>
>
> >2. Theoretical Bounds Lack Practical Guidance: Thm 4.1 does not offer intuition or heuristics for selecting L in new domains.
>
> __Our experimental procedure for selecting the optimal L directly reflects the balance implied by the theoretical bound__.
>
> We first recall Thm 4.1: the performance gap of a PPL agent can be bounded by three terms: the state distribution shift $\delta_\text{dist}$, the quality of the preference dataset $\delta_\text{pref}$, and the training loss $\epsilon$.
>
> From the theoretical bound, __we can expect that PPL will achieve its highest performance when $L$ is balanced (neither too small nor too large).__ We explain it as follows.
>
> - When $L=0$, the first distribution shift term $\delta_{\text{dist}}$ is large, but $\delta_{\text{dist}}$ will drop when $L$ increases.
>
> By definition of $\delta_\text{dist}$ (lines 190–191), $\delta_\text{dist}$ measures the discrepancy between states actually visited by $\pi_n$ and those in the preference dataset; as $L$ grows, the dataset incorporates more predicted future states that will be visited by $\pi_n$, so $\delta_\text{dist}$ decreases.
>
> - When $L=0$, the second error term $\delta_{\text{pref}} = 0$, but $\delta_{\text{pref}}$ will increase as $L$ increases.
>
> By definition (line 198), $\delta_\text{pref}$ measures the error between distributions of the action pairs $(a_h, a_n)$ on current states $s$ and the future states $s'$ in the dataset. As $L$ increases, the predicted states $s'$ drift away from the current states $s$, so $\delta_\text{pref}$ increases.
>
> Therefore, for a well‑trained PPL agent (i.e., training loss $\epsilon$ is small), the optimal selection of $L$ should balance $\delta_\text{dist}$ and $\delta_\text{pref}$.
>
> Experimentally, in MetaDrive, we indeed observe __PPL’s performance rising and then falling as $L$ increases (See Fig. 6), matching our theoretical prediction__. We report it in the following table:
> |MetaDrive |PPL (L = 2) | PPL (L = 3) | PPL (L = 4) |  PPL (L = 5) | PPL (L = 6) | PVP (best baseline) |
> |---|---|---|---|---|---|---|
> |Test Success Rate (trained for 10K steps)| 0.56 | 0.75 | 0.80 | 0.77 | 0.64 | 0.56 |
> |Test Success Rate (trained for 15K steps)| 0.70 | 0.79 | 0.81 | 0.78 | 0.75 | 0.68 |
>
> In conclusion, our method for choosing L in experiments mirrors the trade-off predicted by the theoretical bound.
>
> >3. Missing Comparisons to Key Interactive IL Work: \
> >[1] Liu et al., Robot Learning on the Job: Human‐in‐the‐Loop Autonomy and Learning During Deployment, IJRR 2024 \
> >[2] Liu et al., Multi‐Task Interactive Robot Fleet Learning with Visual World Models, CoRL 2024
>
> Thank you for suggesting additional IIL baselines, and we will include them in the revised manuscript. We implement the method Sirius in [1] given the short rebuttal period, which separates the training data into four categories and uses a Weighted Behavioral Cloning with a high weight given to human interventions and zero weights to pre-intervention states. Following the same setup in Tables 3,4,5 from our paper, we find that __our method PPL outperforms Sirius [1]__ in MetaDrive, Table Wiping, and Nut Assembly tasks:
>
> | MetaDrive | Training: Total Data Usage | Testing: Success Rate | Testing: Route Completion |
> |---|---|---|---|
> | PPL (Ours) | 10K | 0.80 | 0.92 |
> | Sirius [1]     | 10K | 0.42 | 0.73 |
>
> | Table Wiping | Training: Total Data Usage | Testing: Success Rate |
> |---|---|---|
> | PPL (Ours) | 4K | 0.80 |
> | Sirius [1]     | 4K | 0.61 |
>
> | Nut Assembly | Training: Total Data Usage | Testing: Success Rate |
> |---|---|---|
> | PPL (Ours)  | 200K | 0.51 |
> | Sirius [1]     | 200K | 0.34 |
>
> __Questions:__
>
> >1. How did you select the preference horizon L？
>
> Our experiments suggest choosing L up to the number of steps an agent executes in one to two seconds. As shown in Fig. 6, PPL’s performance rises and then falls as L increases, which makes it easy to pinpoint the optimal L: for MetaDrive, the test success rate of PPL climbs from 0.48 at L = 0 to 0.80 at L = 4, then drops to 0.50 by L = 8.
>
> >2. Which trajectory prediction model was employed for RoboSuite experiments, and how sensitive is PPL to its accuracy?
>
> We apply a rollout-based trajectory prediction model for RoboSuite experiments. In the previous question “Dependence on Accurate Predictive Models”, we have conducted experiments showing that __PPL is robust to a noisy prediction model__ up to $\epsilon \leq 0.2$ for Table Wiping and  $\epsilon \leq 0.1$ for Nut Assembly.
>
> >3. Why do different methods use varying amounts of expert data?
>
> The “amount of expert data” is the number of human takeover steps, reflecting how often the expert judged the current state to be unsafe and chose to intervene. Under the same total training steps, methods that make more errors trigger more human takeovers and thus consume more expert data. As Table 1 shows, PPL requires the fewest human interventions (PPL 2.9K vs PVP 4.9K) because its trajectory‑informed policy commits the least errors during training. You can also refer to Sections 1 and 5 of our supplementary demo to see how a human expert decides to take over during training.
>
> > Could you compare all methods using the same data budget?
>
> Yes. As suggested, we compare all methods in MetaDrive with the same human data budget in a new experiment as follows. It shows that __PPL has the best performance with the same expert data usage__.
>
> | Method | Training: Human Data Usage | Testing: Success Rate | Testing: Route Completion |
> |---|---|---|---|
> | PPL (Ours) | 5K | 0.79 | 0.93 |
> | PVP | 5K | 0.46 | 0.71 |
> | IWR | 5K | 0.20 | 0.59 |
> | EIL   | 5K | 0.01 | 0.27 |
> | HACO | 5K | 0.08 | 0.31 |
>
> >4. At what points during training or rollout are expert interventions incorporated into the learning process?
>
> For PPL and all baselines, we interleave data collection and gradient update: after collecting each environment transition, we immediately apply one gradient update. This online update schedule lets the agent use its most recent policy when interacting with the expert, rather than waiting for a large batch of data to accumulate. All baselines follow this schedule to ensure a fair comparison.

---

> ### Author Response · Authors · 2025-08-07
> **Rebuttal and discussion**
>
> Dear Reviewer RuYs,
>
> Can you please kindly come back to check the rebuttal and see if your concerns have been addressed? It seems that all the other reviewers lean toward accepting our work; you are the only one on the negative side, so your opinion matters a lot. We try our best to address any concerns you might have. Thank you.
>
> Authors

---

> > ### Comment · Reviewer_RuYs · 2025-08-08
> >
> > Thanks for your detailed rebuttal. Most of my concerns have been addressed, and I would like to raise my score.

---

### Official Review · Reviewer_5Q4m · 2025-07-02

**Clarity:** 3
**Significance:** 3
**Originality:** 2
**Rating:** 5
**Confidence:** 3

**Summary:**

The technique exploits a trajectory prediction model to forecast the agent's future states. This helps a human supervisor to determine when an intervention is necessary. The proposed approach then converts this intervention into contrastive preference labels over each
predicted state. The resulting dataset provides more information than the corrective demonstrations used by previous approaches.

Results are presented for three tasks (MetaDrive, Table Wiping and Nut Assembly) and comparisons are made to imitation learning baselines (BC, GAIL), confidence-based IIL methods (Ensemble-DAgger and Thifty-Dagger) and human-in-the-loop methods (WR, HACO, EIL, and proxy value propagation). Significant improvements are reported in terms of success rates and reductions in human data usage.

**Questions:**

* Could you justify your selection of the three tasks you use? why only these three?

* minor: section 2. "These methods do not leverage data collected..." - I think this sentence could be reworded for clarity / precision. I think you are trying to say that while they can copy human behaviour they won't suppress undesired actions more generally where humans would anticipate a problem.  Perhaps expand slightly?

**Ethical Concerns:**

["NO or VERY MINOR ethics concerns only"]

**Final Justification:**

I remain positive about this contribution and will retain my rating of 5.

**Limitations:**

yes

**Quality:**

3

**Strengths And Weaknesses:**

strengths:
* a clearly written paper and interesting contribution
* significant improvements over previous approaches both in terms of success rates and human data usage

---

> ### Author Rebuttal · Authors · 2025-07-30
>
> We are grateful for your detailed review and helpful insights. Please find our summarized responses to each of your comments below.
>
> __Questions:__
>
> > 1. Could you justify your selection of the three tasks you use? Why only these three?
>
> Autonomous driving and robotic manipulation are representative real-time tasks where humans can play an essential role in demonstrating and improving the learning process of the AI agents. We selected MetaDrive, Nut Assembly, and Table Wiping to cover diverse challenges: MetaDrive provides a high‑dimensional (259‑D) state space, while Nut Assembly and Table Wiping in RoboSuite feature higher action dimensionality. Nut Assembly, in particular, is among RoboSuite’s hardest tasks due to its extreme sensitivity to small control errors. Even a minor gripper misalignment can prevent the item from being grasped or cause it to drop. In Sections 3 and 4 of the supplementary demos, we show that PPL consistently delivers strong performance across all three.
>
> Moreover, we added additional experiments to demonstrate that __PPL can handle a more complex RGB state space__. We further evaluate PPL in MetaDrive using 320×320 images (three‑frame stacks) as input and a three‑layer CNN encoder within the policy. Even under this high‑dimensional setting, PPL continues to outperform all baselines with 10K training steps:
> | Method | Test Success Rate |
> |--------|--------|
> | PPL (Ours)  | 0.75  |
> | PVP | 0.46  |
> | IWR  | 0.23 |
> | EIL    | 0.08 |
> | HACO  | 0.31 |
>
> > 2. L70-72 “These methods do not leverage data collected by agents or suppress undesired actions likely intervened by humans.” \
>  I think this sentence could be reworded for clarity. \
>  I think you mean that while they can copy human behaviour, they won't suppress undesired actions more generally where humans would anticipate a problem.
>
> We appreciate your suggestions, and your understanding is correct. We rewrite L70-72 as follows:
>
> 1. These IIL baselines (IWR, Thrifty-DAgger, Ensemble-DAgger, HG-DAgger) discard all agent‑collected transitions where no human intervention occurred, wasting valuable self‑exploration data.
>
> 2. Moreover, although prior IIL methods correct the agent’s action at the current state, they do not adjust its actions in the future states, which are potentially more hazardous. Therefore, some dangerous situations may repeatedly appear during training.

---

> > ### Comment · Reviewer_5Q4m · 2025-08-04
> >
> > Thank you for your response to my questions. I have no further questions for you.

---

### Official Review · Reviewer_6PmN · 2025-07-02

**Clarity:** 3
**Significance:** 3
**Originality:** 3
**Rating:** 4
**Confidence:** 3

**Summary:**

The paper proposes  a novel approach to interactive imitation learning (IIL) that enhances learning efficiency by leveraging human interventions more effectively. Traditional methods often require human experts to provide corrective demonstrations reactively, which can be inefficient and burdensome. Instead, the proposed method, Predictive Preference Learning (PPL) uses trajectory prediction models to anticipate future agent actions, allowing experts to intervene proactively. By visualizing predicted trajectories, human supervisors can identify potential failures before they occur, converting these interventions into contrastive preference labels that guide the agent's learning process.

The proposed method significantly reduces the number of interventions needed and decreases the cognitive load on human experts. PPL is evaluated on simulation-based tasks like MetaDrive and Robosuite, demonstrating improved efficiency and generalization compared to existing methods. The paper also includes a theoretical analysis that balances the preference horizon to optimize learning outcomes. Overall, the proposed method offers a promising framework for integrating human preferences into agent training, potentially streamlining the development of more intelligent and adaptable autonomous systems.

**Questions:**

1. The author claims PPL to be a novel IIL method, can author properly defend this claim with substantial proof? The claim written in lines 57-59 doesn't seems sufficient.
2. Can you provide guidance on selecting the preference horizon L? How sensitive is PPL's performance to this parameter?
3. Since LLM/VLM are so popular, did the authors tried to replace/augment human with a LLM? If not, can they discuss how their method might be used with LLM?
4. Since the paper talks about robotic tasks application for the proposed PPL method. However, both experiments MetaDrive and Robosuite shown in the paper are simulation-based tasks. Can author perform real-world robotic task to evaluate the proposed method better? Or at least they can discuss how the proposed method would work in real world and what challenges do you anticipate in transitioning from simulation to real robotic tasks?
5. How does the PPL scale with larger and more complex state spaces, and are there any optimizations planned to improve scalability?
6. What are the computational costs associated with implementing PPL, and how does it compare to other IIL methods in terms of efficiency?

**Ethical Concerns:**

["NO or VERY MINOR ethics concerns only"]

**Final Justification:**

I am satisfied with the discussions during the rebuttal phase and would like to keep my score.

**Limitations:**

Yes the limitations to the proposed method has been listed in the Conclusion section. However, in my opinion the authors should discuss the limitations to PPL in detail in Appendix.

**Quality:**

3

**Strengths And Weaknesses:**

Strengths:
- The proposed method effectively integrates trajectory prediction to inform human interventions, allowing for proactive error correction and reducing the need for constant human oversight.
- With converting interventions into preference labels over predicted trajectories, the proposed approach reduces the cognitive load on human experts and the number of necessary interventions.
- The proposed method is tested on multiple simulation-based benchmarks like MetaDrive and Robosuite, showing improved efficiency and generalization over traditional methods, which supports its practicality and robustness.
- The paper provides a theoretical analysis that explains the balance needed in setting the preference horizon, offering insights into the method's optimization.
- The attached demo video helps to visualize the results better and proposed method effectiveness.

Weakness:
- Since the paper talks about robotic tasks application for the proposed PPL method. However, both experiments MetaDrive and Robosuite shown in the paper are simulation-based tasks. It makes hard for a reviewer to evaluate the paper based on simulation based tasks only. I would encourage the authors to perform some real-world experiments.
- The proposed method assumes that human interventions are always optimal, which might not hold true in practice because of human variability.
- Some key concepts, like "preference horizon" and "contrastive preference labels," could be better explained or illustrated via pictures, enhancing the paper's accessibility.
- The integration of trajectory prediction could introduce additional computational complexity, potentially affecting scalability of the proposed algorithm.
- The paper is not well written and has few unclear statements. Ex. Second statement of the abstract (lines 2-3) is unclear.; There is a typo on line 30 "One line of confidence" -> "One line of research is confidence"; In line 61 what does "neural experts" means? Is it a typo??

---

> ### Author Rebuttal · Authors · 2025-07-30
>
> Thank you for your thorough review and insightful comments. Below, we address each of your questions in detail.
>
> __Other Strengths And Weaknesses:__
>
> >1. I would encourage the authors to perform some real-world experiments.
>
> We appreciate the reviewer’s suggestion regarding real-world experimentation. However, we would like to emphasize that this work focuses on the algorithmic design and theoretical foundations of a new human-in-the-loop machine learning method. As a NeurIPS submission, our primary contribution lies in advancing the core machine learning methodology, which we rigorously validate in simulation environments with real human feedback. We agree that deploying the algorithm in physical systems is an important direction, and we plan to pursue real-world robotic experiments as part of future work.
>
> >2. PPL assumes that human interventions are always optimal.
>
> __We don’t assume human interventions are always optimal.__ We perform the following experiment in MetaDrive to show that PPL is robust to noise and perturbation in the expert. Following Table 3, we use a well-trained PPO expert as a proxy human policy, which achieves a 0.83 success rate. Then, we add Gaussian noises $\mathcal{N}(0, \sigma^2 I)$ to the expert actions, and we evaluate how $\sigma$ affects the test success rate of PPL in MetaDrive with 10K total training steps. The result shows that PPL is robust to a noise $\sigma \leq 0.2$.
> | Noise $\sigma$ in expert policy | Test Success Rate of PPL |
> |--------|--------|
> | 0  | 0.80  |
> | 0.1  | 0.77  |
> | 0.2  | 0.70  |
> | 0.3  | 0.62  |
> | 0.4  | 0.43  |
>
> >3. "preference horizon" and "contrastive preference labels" could be better explained or illustrated via pictures
>
> We have provided the detailed description in Section 1 in our demo in the supplementary material, and we will better explain them and update the pictures in the revised manuscript.
>
> Concretely, the __preference horizon__ L denotes the number of future timesteps over which we solicit human preferences on a predicted trajectory. When an expert intervenes at state $s$ and uses human action $a_h$ to replace the agent action $a_n$, we know $a_h$ is more likely to avert a dangerous outcome not only at $s$ but also at each predicted state $\tilde{s}_1, \cdots, \tilde{s}_L$. Therefore, for each $\tilde{s}_i$ in the horizon, we assign a __contrastive preference label__ $a_h \succ a_n$, indicating that the expert's action is preferred over the agent's.
>
> >4. The integration of trajectory prediction could introduce additional computational complexity
>
> The additional computational cost is minimal compared to the human intervention cost. In our setting in Table 3, we use a very simple trajectory prediction method that can run up to 1000fps on a CPU. Using other off-the-shelf learning-based trajectory prediction models can easily run up to 100fps.
>
> In addition, Table 3 demonstrates that PPL yields a higher success rate (PPL 0.80 vs. 0.56 for the best baseline PVP) and reduces expert takeovers (PPL 1.2 K vs. PVP 2.5 K), so the additional computation cost in fact __saves the human cognitive effort and training time__.
>
> Apart from the rollout-based trajectory predictor, we also developed a faster trajectory extrapolation method by simulating a kinematic bicycle model via forward integration (Lines 320–323), which runs at 3000 fps. As reported in Table 2, this extrapolation approach achieves a 0.78 test success rate in MetaDrive, nearly matching the 0.80 achieved by the rollout predictor. This demonstrates that the choice of predictor has minimal effect on PPL’s performance.
>
> >5. (a) L2-L3 is unclear: “While learning from corrective demonstrations addresses present mistakes, it fails to avert errors in the agent’s subsequent trajectory.“ \
> (b) There is a typo on line 30 "One line of confidence-based IIL works" -> "One line of research is confidence-based IIL works".  \
> (c) What does "neural experts" mean in L61: "We use both neural experts and real human participants"? Is it a typo?
>
> (a) We rewrite L2-L3 as follows: Although prior IIL methods correct the agent’s action at the current state, they do not adjust its actions in the future states, which are potentially more hazardous.
>
> (b) L30 "One line of confidence" is a typo, and we will update it in the revised manuscript.
>
> (c) L61 “neural experts” is not a typo. According to lines 244-246, we also use well-trained PPO policies to approximate human policies in MetaDrive and Robosuite environments, and we report the results in Tables 3,4,5.
>
> __Questions:__
>
> >1. "PPL is a novel IIL method", can the author properly defend this claim with substantial proof?
>
> The core innovation of PPL is the joint integration of trajectory prediction and preference learning into the IIL loop, while no existing IIL baselines (e.g., PVP, IWR, Thrifty‑DAgger) leverage predicted future trajectories to guide learning. Moreover, unlike traditional preference‑based RL methods such as RLHF and DPO that train on static, offline human‑annotated datasets, PPL continuously learns human preferences in an online setting from real‑time expert interventions.
>
> >2. Can you provide guidance on selecting the preference horizon L? How sensitive is PPL's performance to this parameter?
>
> Our experiments suggest choosing L up to the number of steps that an agent executes in one to two seconds. In addition, __PPL’s performance rises and then falls as L increases__, which makes it easy to pinpoint the optimal L. As shown in Fig. 6, for MetaDrive, the success rate climbs from 0.48 at L = 0 to 0.80 at L = 4, then drops to 0.50 by L = 8.
>
> __PPL is not very sensitive to L.__ We find that __PPL with any L between 2 and 6 outperforms all baselines__ in MetaDrive. The following table reports the test success rate of PPL with different values of L with 10K and 15K total training data usage.
> | |PPL (L = 2) | PPL (L = 3) | PPL (L = 4) |  PPL (L = 5) | PPL (L = 6) | PVP (best baseline) |
> |---|---|---|---|---|---|---|
> |Test Success Rate (trained for 10K steps)| 0.56 | 0.75 | 0.80 | 0.77 | 0.64 | 0.56 |
> |Test Success Rate (trained for 15K steps)| 0.70 | 0.79 | 0.81 | 0.78 | 0.75 | 0.68 |
>
> >3. Did the authors try to replace/augment humans with an LLM? Discuss how their method might be used with LLM.
>
> Although we have not yet tested PPL with an LLM/VLM as the expert, our PPL framework supports such integration. To incorporate the VLM as an expert in PPL's setting, we can feed the agent's predicted future trajectory (rendered as front‑view images) to the VLM and let it decide whether to intervene. Upon takeover, the model would choose its preferred action from a small set of predefined options, and the low-level policy will be trained to follow the VLM's preferences. In future work, fine-tuning the VLM on intervention data may further improve its guidance.
>
> >4. How would the proposed method work in the real world, and what challenges do you anticipate in transitioning from simulation to real robotic tasks?
>
> We anticipate two main challenges: accurate trajectory predictors are harder to obtain, and human takeovers are more expensive. Additionally, distribution shifts in novel environments may degrade predictor performance. We plan to tackle the real-world deployment issues in our future work.
>
> >5. How does the PPL scale with larger and more complex state spaces, and are there any optimizations planned to improve scalability?
>
> We added experiments in MetaDrive to demonstrate that __PPL can handle a more complex RGB state space__. We further evaluate PPL in MetaDrive using 320×320 images (three‑frame stacks) as input and a three‑layer CNN encoder within the policy. Even under this high‑dimensional setting, PPL continues to outperform all baselines with 10K training steps:
> | Method | Test Success Rate |
> |--------|--------|
> | PPL (Ours)  | 0.75  |
> | PVP | 0.46  |
> | IWR  | 0.23 |
> | EIL    | 0.08 |
> | HACO  | 0.31 |
>
> For tasks with even more demanding observation spaces, one can employ pretrained encoders to extract compact feature embeddings and reduce computation.
>
> >6. What are the computational costs associated with implementing PPL compared to other IIL methods?
>
> In our main experiments (Tables 1 and 3–5), PPL’s computation runs at ~1,000 fps, comparable to the baseline methods (Thrifty‑DAgger, PVP, IWR, etc.). Moreover, Table 3 demonstrates that PPL yields a higher success rate (PPL 0.80 vs. PVP 0.56) and reduces expert takeovers (PPL 1.2 K vs. PVP 2.5 K), demonstrating that PPL actually lowers human cognitive load and shortens total training time.

---

> > ### Comment · Reviewer_6PmN · 2025-08-06
> >
> > Thank you so much for your responses!  Mostly of my concerns have been addressed, and I appreciate the clarity and depth you provided. I look forward to seeing how your work evolves, particularly with real-world applications and further enhancements to the methodology. One of the things I am still not clear on is Q6 about computational complexity?

---

> ### Author Response · Authors · 2025-08-06
> **Detailed explanation of Q6 about computational complexity**
>
> Thanks for your response! We highlight that PPL adds one extra L-step trajectory prediction module, but this overhead of computation is minor in practice.
>
> First, our L-step rollout predictor, or the faster extrapolation method, consumes less than 5% of the end-to-end training time and runs at over 1000 fps on a CPU. As shown in Section 1 of our demo, each time the vehicle moves or a human takeover occurs, the predicted trajectory appears instantly, and no waiting is required.
>
> Second, the training computation cost scales with the total environment steps. In Table 3, PPL reaches a 0.80 success rate in under 10K steps, whereas the best baseline PVP only achieves 0.56 in the same 10K budget. Also, PVP requires at least 30K steps to approach 0.80. Thus, PPL has a much shorter training time and computation cost.
>
> Finally, in our human-in-the-loop experiments (See Section 1 in our demo), the dominant computational expense is image rendering at a 60 fps, which matches the maximum rate at which humans can accurately monitor the agent and provide interventions.
>
> In conclusion, PPL's trajectory prediction module adds negligible overhead while resulting in significant reductions in training steps and computation cost, making PPL highly efficient in practice. PPL also requires fewer expert takeovers during training (PPL 1.2 K vs. PVP 2.5 K), which is more human-friendly. We also intend to implement PPL on physical robotic platforms in our future work.

---

### Comment · Area_Chair_XW3j · 2025-08-04
**Please engage with the authors responses**

The authors have provided detailed responses. Please engage with their responses and take advantage of the short window that remains to ask for any additional clarifications if needed.

---

### Comment · Area_Chair_XW3j · 2025-08-04
**Please engage with the authors responses**

The authors have provided detailed responses. Please engage with their responses and take advantage of the short window that remains to ask for any additional clarifications if needed.

---

### Comment · Area_Chair_XW3j · 2025-08-06

I'm concerned about the lack of engagement with the author responses on this paper. Please be sure to read the author responses, acknowledge you have done so, and use the opportunity to seek any additional clarifications from the authors while the brief window to do so remains open.

Thanks!

---

### Note · Authors · 2025-08-15

We thank all the reviewers for their careful reading and valuable suggestions.
Below, we briefly summarize the main points addressed in our rebuttal:

1. **Trajectory prediction's efficiency and PPL's robustness**: Predictive Preference Learning (PPL) uses a lightweight trajectory predictor to guide human interventions and learn human preference, **reducing human takeover costs and shortening training time**. Our predictor runs at up to 1000 fps on CPU, and our demo shows that training PPL takes only about 10 minutes, illustrating its efficiency and human-friendliness. We also show **PPL's robustness to predictor noise**: in our quantitative tests, PPL tolerates disturbances up to $\epsilon \le 0.2$, where $\epsilon$ is the ratio between the noise norm and the predictor output norm.
2. **Theory guides the choice of optimal preference horizon $L$**:  Empirically, PPL’s performance rises then falls as $L$ increases, and Theorem 4.1 indicates that the optimal $L$ should balance two error terms that change in opposite directions with $L$ (see our response to Reviewer 6PmN). This implies that **the bound provides practical guidance for selecting $L$**. We also verified the theoretical condition on $\beta$ and found that a small $\beta$ is helpful in practice.
3. **Baseline comparisons**: PPL outperforms all baselines in robotic manipulation and autonomous driving, including the additional baseline Sirius (see our response to Reviewer RuYs). From our additional experiments, PPL works for **high-dimensional RGB state spaces** in driving tasks, and it remains effective in Robosuite environments whose dynamics are sensitive to control errors. **PPL also remains effective when the expert demonstrations are noisy or imperfect.**

In summary, PPL is a novel interactive imitation learning method that **combines trajectory prediction with online preference learning** to achieve faster training and fewer required human samples. We evaluate PPL on manipulation and driving tasks with **real human participants** and provide a theoretical analysis that upper-bounds the performance gap.

We again thank the reviewers and AC for their insightful feedback.

---

### Decision · Program_Chairs · 2025-09-17

**Decision:**

Accept (spotlight)

**Comment:**

The authors did a good job of addressing numerous concerns raised by the reviewers, ultimately bringing all reviewers into a positive view of the paper. The concerns raised by the reviewers appear, generally, to be reasonable questions that a typical reader might have about the paper, so it will be important for the authors to revise their paper in way that anticipates and answers these concerns for future readers.